# Optimal Design for Human Preference Elicitation

**Subhojyoti Mukherjee**
University of Wisconsin-Madison*
smukherjee27@wisc.edu

**Anusha Lalitha**
AWS AI Labs

**Kousha Kalantari**
AWS AI Labs

**Aniket Deshmukh**
AWS AI Labs

**Ge Liu**
UIUC*

**Yifei Ma**
AWS AI Labs

**Branislav Kveton**
Adobe Research*

## Abstract

Learning of preference models from human feedback has been central to recent advances in artificial intelligence. Motivated by the cost of obtaining high-quality human annotations, we study efficient human preference elicitation for learning preference models. The key idea in our work is to generalize optimal designs, an approach to computing optimal information-gathering policies, to lists of items that represent potential questions with answers. The policy is a distribution over the lists and we elicit preferences from them proportionally to their probabilities. To show the generality of our ideas, we study both absolute and ranking feedback models on items in the list. We design efficient algorithms for both and analyze them. Finally, we demonstrate that our algorithms are practical by evaluating them on existing question-answering problems.

## 1 Introduction

*Reinforcement learning from human feedback (RLHF)* has been effective in aligning and fine-tuning *large language models (LLMs)* [68, 36, 15, 77, 39]. The main difference from classic *reinforcement learning (RL)* [81] is that the agent learns from human feedback, which is expressed as preferences for different potential choices [2, 49, 71, 10, 90]. The human feedback allows LLMs to be adapted beyond the distribution of data that was used for their pre-training and generate answers that are more preferred by humans [15]. The feedback can be incorporated by learning a preference model. When the human decides between two choices, the *Bradley-Terry-Luce (BTL)* model [12] can be used. For multiple choices, the *Plackett-Luce (PL)* model [65, 54] can be adopted. A good preference model should correctly rank answers to many potential questions. Therefore, learning of a good preference model can be viewed as learning to rank, and we adopt this view in this work. Learning to rank has been studied extensively in both offline [14] and online [67, 43, 83, 80, 46] settings.

To effectively learn preference models, we study efficient methods for human preference elicitation. We formalize this problem as follows. We have a set of $L$ *lists* representing *questions*, each with $K$ *items* representing *answers*. The objective of the agent is to learn to rank all items in all lists. The agent can query humans for feedback. Each query is a question with $K$ answers represented as a list. The human provides feedback on it. We study two feedback models: absolute and ranking. In the absolute feedback model, a human provides noisy feedback for each item in the list. This setting is motivated by how annotators assign relevance judgments in search [30, 57]. The ranking feedback is motivated by learning reward models in RLHF [68, 36, 15, 77, 39]. In this model, a human ranks all items in the list according to their preferences. While $K = 2$ is arguably the most common case, we study $K \geq 2$ for the sake of generality and allowing a higher-capacity communication channel with the human [102]. The agent has a budget for the number of queries. To learn efficiently within the

---

*The work was done at AWS AI Labs.

38th Conference on Neural Information Processing Systems (NeurIPS 2024).

budget, it needs to elicit preferences from the most informative lists, which allows it to learn to rank all other lists. Our main contribution is an efficient algorithm for computing the distribution of the most informative lists.

Our work touches on many topics. Learning of reward models from human feedback is at the center of RLHF [62] and its recent popularity has led to major theory developments, including analyses of regret minimization in RLHF [16, 87, 90, 91, 61, 75]. These works propose and analyze adaptive algorithms that interact with the environment to learn highly-rewarding policies. Such policies are usually hard to deploy in practice because they may harm user experience due to over-exploration [22, 82]. Therefore, Zhu et al. [102] studied RLHF from ranking feedback in the offline setting with a fixed dataset. We study how to collect an *informative dataset for offline learning to rank* with both absolute and ranking feedback. We approach this problem as an optimal design, a methodology for computing optimal information-gathering policies [66, 24]. The policies are non-adaptive and thus can be precomputed, which is one of their advantages. The main technical contribution of this work is a matrix generalization of the Kiefer-Wolfowitz theorem [41], which allows us to formulate optimal designs for ranked lists and solve them efficiently. Optimal designs have become a standard tool in exploration [45, 37, 38, 58, 33] and adaptive algorithms can be obtained by combining them with elimination. Therefore, optimal designs are a natural stepping stone to other solutions.

We make the following contributions:

1. We develop a novel approach for human preference elicitation. The key idea is to generalize the Kiefer-Wolfowitz theorem [41] to matrices (Section 3), which then allows us to compute information-gathering policies for ranked lists.

2. We propose an algorithm that uses an optimal design to collect absolute human feedback (Section 4.1), where a human provides noisy feedback for each item in the queried list. A least-squares estimator is then used to learn a preference model. The resulting algorithm is both computationally and statistically efficient. We bound its prediction error (Section 4.2) and ranking loss (Section 4.3), and show that both decrease with the sample size.

3. We propose an algorithm that uses an optimal design to collect ranking human feedback (Section 5.1), where a human ranks all items in the list according to their preferences. An estimator of Zhu et al. [102] is then used to learn a preference model. Our approach is both computationally and statistically efficient, and we bound its prediction error (Section 5.2) and ranking loss (Section 5.3). These results mimic the absolute feedback setting and show the generality of our framework.

4. We compare our algorithms to multiple baselines in several experiments. We observe that the algorithms achieve a lower ranking loss than the baselines.

## 2 Setting

**Notation:** Let $[K] = \{1, \ldots, K\}$. Let $\Delta^L$ be the probability simplex over $[L]$. For any distribution $\pi \in \Delta^L$, we get $\sum_{i=1}^{L} \pi(i) = 1$. Let $\Pi_2(K) = \{(j, k) \in [K]^2 : j < k\}$ be the set of all pairs over $[K]$ where the first entry is lower than the second one. Let $\|\mathbf{x}\|_{\mathbf{A}}^2 = \mathbf{x}^\top \mathbf{A}\mathbf{x}$ for any positive-definite $\mathbf{A} \in \mathbb{R}^{d \times d}$ and $\mathbf{x} \in \mathbb{R}^d$. We use $\tilde{O}$ for the big-O notation up to logarithmic factors. Specifically, for any function $f$, we write $\tilde{O}(f(n))$ if it is $O(f(n) \log^k f(n))$ for some $k > 0$. Let $\mathrm{supp}\,(\pi)$ be the support of distribution $\pi$ or a random variable.

**Setup:** We learn to rank $L$ lists, each with $K$ items. An item $k \in [K]$ in list $i \in [L]$ is represented by a feature vector $\mathbf{x}_{i,k} \in \mathcal{X}$, where $\mathcal{X} \subseteq \mathbb{R}^d$ is the set of feature vectors. The relevance of items is given by their mean rewards. The mean reward of item $k$ in list $i$ is $\mathbf{x}_{i,k}^\top \boldsymbol{\theta}_*$, where $\boldsymbol{\theta}_* \in \mathbb{R}^d$ is an unknown parameter. Without loss of generality, we assume that the original order of the items is optimal, $\mathbf{x}_{i,j}^\top \boldsymbol{\theta}_* > \mathbf{x}_{i,k}^\top \boldsymbol{\theta}_*$ for any $j < k$ and list $i$. The agent does not know it. The agent interacts with humans for $n$ rounds. At round $t$, it selects a list $I_t$ and the human provides stochastic feedback on it. Our goal is to design a policy for selecting the lists such that the agent learns the optimal order of all items in all lists after $n$ rounds.

**Feedback model:** We study two models of human feedback, absolute and ranking:

(1) In the *absolute feedback model*, the human provides a reward for each item in list $I_t$ chosen by the agent. Specifically, the agent observes noisy rewards

$$y_{t,k} = \mathbf{x}_{I_t,k}^\top \boldsymbol{\theta}_* + \eta_{t,k} \,, \tag{1}$$

for all $k \in [K]$ in list $I_t$, where $\eta_{t,k}$ is independent zero-mean 1-sub-Gaussian noise. This feedback is stochastic and similar to that in the document-based click model [19].

(2) In the *ranking feedback model*, the human orders all items in list $I_t$ selected by the agent. The feedback is a permutation $\sigma_t : [K] \to [K]$, where $\sigma_t(k)$ is the index of the $k$-th ranked item. The probability that this permutation is generated is

$$p(\sigma_t) = \prod_{k=1}^K \frac{\exp[\mathbf{x}_{I_t,\sigma_t(k)}^\top \boldsymbol{\theta}_*]}{\sum_{j=k}^K \exp[\mathbf{x}_{I_t,\sigma_t(j)}^\top \boldsymbol{\theta}_*]} \,. \tag{2}$$

Simply put, items with higher mean rewards are more preferred by humans and hence more likely to be ranked higher. This feedback model is known as the *Plackett-Luce (PL)* model [65, 54, 102], and it is a standard assumption when learning values of individual choices from relative feedback. Since the feedback at round $t$ is with independent noise, in both (1) and (2), any list can be observed multiple times and we do need to assume that $n \le L$.

**Objective:** At the end of round $n$, the agent outputs a permutation $\hat{\sigma}_{n,i} : [K] \to [K]$ for all lists $i \in [L]$, where $\hat{\sigma}_{n,i}(k)$ is the index of the $k$-th ranked item in list $i$. We measure the quality of the solution by the *ranking loss* after $n$ rounds, which we define as

$$\mathrm{R}_n = \sum_{i=1}^L \sum_{j=1}^K \sum_{k=j+1}^K \mathbb{1}\{\hat{\sigma}_{n,i}(j) > \hat{\sigma}_{n,i}(k)\} \,. \tag{3}$$

The loss is the number of incorrectly ordered pairs of items in permutation $\hat{\sigma}_{n,i}$, summed over all lists $i \in [L]$. It can also be viewed as the Kendall tau rank distance [40] between the optimal order of items in all lists and that according to $\hat{\sigma}_{n,i}$. We note that other ranking metrics exist, such as the *normalized discounted cumulative gain (NDCG)* [86] and *mean reciprocal rank (MRR)* [85]. Our work can be extended to them and we leave this for future work.

The two closest related works are Mehta et al. [56] and Das et al. [20]. They proposed algorithms for learning to rank $L$ pairs of items from pairwise feedback. Their optimized metric is the maximum gap over the $L$ pairs. We learn to rank $L$ lists of $K$ items from $K$-way ranking feedback. We bound the maximum prediction error, which is a similar metric to the prior works, and the ranking loss in (3), which is novel. Our setting is related to other bandit settings as follows. Due to the budget $n$, it is similar to fixed-budget *best arm identification (BAI)* [13, 5, 6, 95]. The main difference is that we do not want to identify the best arm. We want to sort $L$ lists of $K$ items. Online learning to rank has also been studied extensively [67, 43, 105, 53, 44]. We do not minimize cumulative regret or try to identify the best arm. A more detailed comparison is in Appendix D.

We introduce optimal designs [66, 24] next. This allows us to minimize the expected ranking loss within a budget of $n$ rounds efficiently.

## 3   Optimal Design and Matrix Kiefer-Wolfowitz

This section introduces a unified approach to human preference elicitation from both absolute and ranking feedback. First, we note that to learn the optimal order of items in all lists, the agent has to estimate the unknown model parameter $\boldsymbol{\theta}_*$ well. In this work, the agent uses a *maximum-likelihood estimator (MLE)* to obtain an estimate $\hat{\boldsymbol{\theta}}_n$ of $\boldsymbol{\theta}_*$. After that, it orders the items in all lists according to their estimated mean rewards $\mathbf{x}_{i,k}^\top \hat{\boldsymbol{\theta}}_n$ in descending order, which defines the permutation $\hat{\sigma}_{n,i}$. If $\hat{\boldsymbol{\theta}}_n$ minimized the prediction error $(\mathbf{x}_{i,k}^\top (\hat{\boldsymbol{\theta}}_n - \boldsymbol{\theta}_*))^2$ over all items $k \in [K]$ in list $i$, the permutation $\hat{\sigma}_{n,i}$ would be closer to the optimal order. Moreover, if $\hat{\boldsymbol{\theta}}_n$ minimized the maximum error over all lists, all permutations would be closer and the ranking loss in (3) would be minimized. This is why we focus on minimizing the *maximum prediction error*

$$\max_{i \in [L]} \sum_{\mathbf{a} \in \mathbf{A}_i} (\mathbf{a}^\top (\hat{\boldsymbol{\theta}}_n - \boldsymbol{\theta}_*))^2 = \max_{i \in [L]} \mathrm{tr}(\mathbf{A}_i^\top (\hat{\boldsymbol{\theta}}_n - \boldsymbol{\theta}_*)(\hat{\boldsymbol{\theta}}_n - \boldsymbol{\theta}_*)^\top \mathbf{A}_i) \,, \tag{4}$$

where $\mathbf{A}_i$ is a matrix representing list $i$ and $\mathbf{a} \in \mathbf{A}_i$ is a column in it. In the absolute feedback model, the columns of $\mathbf{A}_i$ are feature vectors of items in list $i$ (Section 4.1). In the ranking feedback model, the columns of $\mathbf{A}_i$ are the differences of feature vectors of items in list $i$ (Section 5.1). Therefore, $\mathbf{A}_i$ depends on the type of human feedback. In fact, as we show later, it is dictated by the covariance of $\hat{\boldsymbol{\theta}}_n$ in the corresponding human feedback model. We note that the objective in (4) is worst-case over lists and that other alternatives, such as $\frac{1}{L} \sum_{i=1}^{L} \sum_{\mathbf{a} \in \mathbf{A}_i} (\mathbf{a}^\top (\hat{\boldsymbol{\theta}}_n - \boldsymbol{\theta}_*))^2$, may be possible. We leave this for future work.

We prove in Sections 4 and 5 that the agent can minimize the maximum prediction error in (4) and the ranking loss in (3) by sampling from a fixed distribution $\pi_* \in \Delta^L$. That is, the probability of selecting list $i$ at round $t$ is $\mathbb{P}\,(I_t = i) = \pi_*(i)$. The distribution $\pi_*$ is a minimizer of

$$g(\pi) = \max_{i \in [L]} \operatorname{tr}(\mathbf{A}_i^\top \mathbf{V}_\pi^{-1} \mathbf{A}_i)\,, \tag{5}$$

where $\mathbf{V}_\pi = \sum_{i=1}^{L} \pi(i) \mathbf{A}_i \mathbf{A}_i^\top$ is a *design matrix*. The *optimal design* aims to find the distribution $\pi_*$. Since (5) does not depend on the received feedback, our algorithms are not adaptive.

The problem of finding $\pi_*$ that minimizes (5) is called the *G-optimal design* [45]. The minimum of (5) and the support of $\pi_*$ are characterized by the Kiefer-Wolfowitz theorem [41, 45]. The original theorem is for least-squares regression, where $\mathbf{A}_i$ are feature vectors. At a high level, it says that the smallest ellipsoid that covers all feature vectors has the minimum volume, and in this way relates the minimization of (5) to maximizing $\log \det(\mathbf{V}_\pi)$. We generalize this claim to lists, where $\mathbf{A}_i$ is a matrix of feature vectors representing list $i$. This generalization allows us to go from a design over feature vectors to a design over lists represented by matrices.

**Theorem 1** (Matrix Kiefer-Wolfowitz). *Let $M \geq 1$ be an integer and $\mathbf{A}_1, \ldots, \mathbf{A}_L \in \mathbb{R}^{d \times M}$ be $L$ matrices whose column space spans $\mathbb{R}^d$. Then the following claims are equivalent:*

*(a) $\pi_*$ is a minimizer of $g(\pi)$ in (5).*

*(b) $\pi_*$ is a maximizer of $f(\pi) = \log \det(\mathbf{V}_\pi)$.*

*(c) $g(\pi_*) = d$.*

*Furthermore, there exists a minimizer $\pi_*$ of $g(\pi)$ such that $|\operatorname{supp}(\pi_*)| \leq d(d+1)/2$.*

*Proof.* We generalize the proof of the Kiefer-Wolfowitz theorem in Lattimore and Szepesvari [45]. The key observation is that even if $\mathbf{A}_i$ is a matrix and not a vector, the design matrix $\mathbf{V}_\pi$ is positive definite. Using this structure, we establish the key facts used in the original proof. First, we show that $\nabla f(\pi) = (\operatorname{tr}(\mathbf{A}_i^\top \mathbf{V}_\pi^{-1} \mathbf{A}_i))_{i=1}^{L}$ is the gradient of $f(\pi)$ with respect to $\pi$. In addition, we prove that $g(\pi) \geq \sum_{i=1}^{L} \pi(i) \operatorname{tr}(\mathbf{A}_i^\top \mathbf{V}_\pi^{-1} \mathbf{A}_i) = d$. The complete proof is in Appendix A.1. $\qquad\square$

From the equivalence in Theorem 1, it follows that the agent should solve the optimal design

$$\pi_* = \operatorname*{arg\,max}_{\pi \in \Delta^L} f(\pi) = \operatorname*{arg\,max}_{\pi \in \Delta^L} \log \det(\mathbf{V}_\pi) \tag{6}$$

and sample according to $\pi_*$ to minimize the maximum prediction error in (4). Note that the optimal design over lists in (6) is different from the one over vectors [45]. As an example, suppose that we have 4 feature vectors $\{\mathbf{x}_i\}_{i \in [4]}$ and two lists: $\mathbf{A}_1 = (\mathbf{x}_1, \mathbf{x}_2)$ and $\mathbf{A}_2 = (\mathbf{x}_3, \mathbf{x}_4)$. The list design is over 2 variables (lists) while the vector design is over 4 variables (vectors). The list design can also be viewed as a constrained vector design, where $(\mathbf{x}_1, \mathbf{x}_2)$ and $(\mathbf{x}_3, \mathbf{x}_4)$ are observed together with the same probability.

The optimization problem in (6) is convex and thus easy to solve. When the number of lists is large, the Frank-Wolfe algorithm [59, 32] can be used, which solves convex optimization problems with linear constraints as a sequence of linear programs. We use CVXPY [21] to compute the optimal design. We report its computation time, as a function of the number of lists $L$, in Appendix E. The computation time scales roughly linearly with the number of lists $L$. In the following sections, we employ Theorem 1 to bound the maximum prediction error and ranking loss for both absolute and ranking feedback.

| **Algorithm 1** Dope for absolute feedback. | **Algorithm 2** Dope for ranking feedback. |
|---|---|
| 1: **for** $i = 1, \ldots, L$ **do** | 1: **for** $i = 1, \ldots, L$ **do** |
| 2:     $\mathbf{A}_i \leftarrow [\mathbf{x}_{i,k}]_{k \in [K]}$ | 2:     **for** $(j, k) \in \Pi_2(K)$ **do** |
| 3: $\mathbf{V}_\pi \leftarrow \sum_{i=1}^L \pi(i) \mathbf{A}_i \mathbf{A}_i^\top$ | 3:       $\mathbf{z}_{i,j,k} \leftarrow \mathbf{x}_{i,j} - \mathbf{x}_{i,k}$ |
| 4: $\pi_* \leftarrow \arg\max_{\pi \in \Delta^L} \log \det(\mathbf{V}_\pi)$ | 4:     $\mathbf{A}_i \leftarrow [\mathbf{z}_{i,j,k}]_{(j,k) \in \Pi_2(K)}$ |
| 5: **for** $t = 1, \ldots, n$ **do** | 5: $\mathbf{V}_\pi \leftarrow \sum_{i=1}^L \pi(i) \mathbf{A}_i \mathbf{A}_i^\top$ |
| 6:     Sample $I_t \sim \pi_*$ | 6: $\pi_* \leftarrow \arg\max_{\pi \in \Delta^L} \log \det(\mathbf{V}_\pi)$ |
| 7:     **for** $k = 1, \ldots, K$ **do** | 7: **for** $t = 1, \ldots, n$ **do** |
| 8:       Observe $y_{t,k}$ in (1) | 8:     Sample $I_t \sim \pi_*$ |
| 9: Compute $\hat{\boldsymbol{\theta}}_n$ in (7) | 9:     Observe $\sigma_t$ in (2) |
| 10: **for** $i = 1, \ldots, L$ **do** | 10: Compute $\hat{\boldsymbol{\theta}}_n$ in (10) |
| 11:     Set $\hat{\sigma}_{n,i}(k)$ to the index of the item with the $k$-th highest $\mathbf{x}_{i,\ell}^\top \hat{\boldsymbol{\theta}}_n$ in list $i$ | 11: **for** $i = 1, \ldots, L$ **do** |
| 12: **Output:** Permutation $\hat{\sigma}_{n,i}$ for all $i \in [L]$ | 12:     Set $\hat{\sigma}_{n,i}(k)$ to the index of the item with the $k$-th highest $\mathbf{x}_{i,\ell}^\top \hat{\boldsymbol{\theta}}_n$ in list $i$ |
| | 13: **Output:** Permutation $\hat{\sigma}_{n,i}$ for all $i \in [L]$ |

## 4 Learning with Absolute Feedback

This section is organized as follows. In Section 4.1, we present an algorithm for human preference elicitation under absolute feedback. We bound its prediction error in Section 4.2 and its ranking loss in Section 4.3.

### 4.1 Algorithm Dope

Our algorithm for absolute feedback is called **D**-**o**ptimal **p**reference **e**licitation (Dope). It has four main parts. First, we solve the optimal design in (6) to get a data logging policy $\pi_*$. The matrix for list $i$ is $\mathbf{A}_i = [\mathbf{x}_{i,k}]_{k \in [K]} \in \mathbb{R}^{d \times K}$, where $\mathbf{x}_{i,k}$ is the feature vector of item $k$ in list $i$. Second, we collect human feedback for $n$ rounds. At round $t \in [n]$, we sample a list $I_t \sim \pi_*$ and then observe $y_{t,k}$ for all $k \in [K]$, as defined in (1). Third, we estimate the model parameter using least squares

$$\hat{\boldsymbol{\theta}}_n = \bar{\boldsymbol{\Sigma}}_n^{-1} \sum_{t=1}^n \sum_{k=1}^K \mathbf{x}_{I_t,k} y_{t,k} \,. \tag{7}$$

The normalized and unnormalized covariance matrices corresponding to the estimate are

$$\boldsymbol{\Sigma}_n = \frac{1}{n} \bar{\boldsymbol{\Sigma}}_n \,, \quad \bar{\boldsymbol{\Sigma}}_n = \sum_{t=1}^n \sum_{k=1}^K \mathbf{x}_{I_t,k} \mathbf{x}_{I_t,k}^\top \,, \tag{8}$$

respectively. Finally, we sort the items in all lists $i$ according to their estimated mean rewards $\mathbf{x}_{i,k}^\top \hat{\boldsymbol{\theta}}_n$ in descending order, to obtain the permutation $\hat{\sigma}_{n,i}$. The pseudo-code of Dope is in Algorithm 1.

The estimator (7) is the same as in *ordinary least squares (OLS)*, because each observed list can be treated as $K$ independent observations. The matrix for list $i$, $\mathbf{A}_i$, can be related to the inner sum in (8) through $\text{tr}(\mathbf{A}_i \mathbf{A}_i^\top) = \sum_{k=1}^K \mathbf{x}_{i,k} \mathbf{x}_{i,k}^\top$. Therefore, our algorithm collects data for a least-squares estimator by optimizing its covariance [45, 33].

### 4.2 Maximum Prediction Error Under Absolute Feedback

In this section, we bound the maximum prediction error of Dope under absolute feedback. We start with a lemma that uses the optimal design $\pi_*$ to bound $\max_{i \in [L]} \sum_{\mathbf{a} \in \mathbf{A}_i} \|\mathbf{a}\|_{\bar{\boldsymbol{\Sigma}}_n^{-1}}^2$.

**Lemma 2.** *Let $\pi_*$ be the optimal design in (6). Fix budget $n$ and let each allocation $n\pi_*(i)$ be an integer. Then $\max_{i \in [L]} \sum_{\mathbf{a} \in \mathbf{A}_i} \|\mathbf{a}\|_{\bar{\boldsymbol{\Sigma}}_n^{-1}}^2 = d/n$.*

The lemma is proved in Appendix A.2. Since all $n\pi_*(i)$ are integers, we note that $\bar{\boldsymbol{\Sigma}}_n$ must be full rank and invertible. Note that the assumption of all $n\pi_*(i)$ being integers does not require $n \geq L$.

This is because $\pi_*(i)$ has at most $d(d + 1)/2$ non-zero entries (Theorem 1). This is independent of the number of lists $L$, which could also be infinite (Chapter 21.1 in Lattimore and Szepesvari [45]). The integer condition can be also relaxed by rounding non-zero entries of $n\pi_*(i)$ up to the closest integer. This clearly yields an integer allocation of size at most $n + d(d + 1)/2$. All claims in our work would hold for any $\pi_*$ and this allocation. With Lemma 2 in hand, the maximum prediction error is bounded as follows.

**Theorem 3** (Maximum prediction error). *With probability at least $1 - \delta$, the maximum prediction error after $n$ rounds is*

$$\max_{i \in [L]} \mathrm{tr}(\mathbf{A}_i^\top (\hat{\boldsymbol{\theta}}_n - \boldsymbol{\theta}_*)(\hat{\boldsymbol{\theta}}_n - \boldsymbol{\theta}_*)^\top \mathbf{A}_i) = O\left(\frac{d^2 + d\log(1/\delta)}{n}\right) \ .$$

The theorem is proved in Appendix A.3. As in Lemma 2, we assume that each allocation $n\pi_*(i)$ is an integer. If the allocations were not integers, rounding errors would arise and need to be bounded [66, 25, 37]. At a high level, our bound would be multiplied by $1 + \beta$ for some $\beta > 0$ (Chapter 21 in Lattimore and Szepesvari [45]). We omit this factor in our proofs to simplify them.

Theorem 3 says that the maximum prediction error is $\tilde{O}(d^2/n)$. Note that this rate cannot be attained trivially, for instance by uniform sampling. To see this, consider the following example. Take $K = 2$. Let $\mathbf{x}_{i,1} = (1, 0, 0)$ for $i \in [L - 1]$ and $\mathbf{x}_{L,1} = (0, 1, 0)$, and $\mathbf{x}_{i,2} = (0, 0, 1)$ for all $i \in [L]$. In this case, the minimum eigenvalue of $\bar{\boldsymbol{\Sigma}}_n$ is $n/L$ in expectation, because only one item in list $L$ provides information about the second feature, $\mathbf{x}_{L,1} = (0, 1, 0)$. Following the proof of Theorem 3, we would get a rate of $\tilde{O}(dL/n)$. Prior works on optimal designs also made similar observations [78].

The rate in Theorem 3 is the same as in linear models [45]. Specifically, by the Cauchy-Schwarz inequality, we would get

$$(\mathbf{x}^\top (\hat{\boldsymbol{\theta}}_n - \boldsymbol{\theta}_*))^2 \leq \|\hat{\boldsymbol{\theta}}_n - \boldsymbol{\theta}_*\|_{\bar{\boldsymbol{\Sigma}}_n}^2 \|\mathbf{x}\|_{\bar{\boldsymbol{\Sigma}}_n^{-1}}^2 = \tilde{O}(d)\,\tilde{O}(d/n) = \tilde{O}(d^2/n)$$

with a high probability, where $\boldsymbol{\theta}_*$, $\hat{\boldsymbol{\theta}}_n$, and $\bar{\boldsymbol{\Sigma}}_n$ are the analogous linear model quantities. This bound holds for infinitely many feature vectors. It can be tightened to $\tilde{O}(d/n)$ for a finite number of feature vectors, where $\tilde{O}$ hides the logarithm of the number of feature vectors. This can be proved using a union bound over (20.3) in Chapter 20 of Lattimore and Szepesvari [45].

## 4.3 Ranking Loss Under Absolute Feedback

In this section, we bound the expected ranking loss under absolute feedback. Recall from Section 2 that the original order of items in each list is optimal. With this in mind, the *gap* between the mean rewards of items $j$ and $k$ in list $i$ is $\Delta_{i,j,k} = (\mathbf{x}_{i,j} - \mathbf{x}_{i,k})^\top \boldsymbol{\theta}_*$, for any $i \in [L]$ and $(j, k) \in \Pi_2(K)$.

**Theorem 4** (Ranking loss). *The expected ranking loss after $n$ rounds is bounded as*

$$\mathbb{E}[\mathrm{R}_n] \leq 2 \sum_{i=1}^{L} \sum_{j=1}^{K} \sum_{k=j+1}^{K} \exp\left[-\frac{\Delta_{i,j,k}^2 n}{8d}\right] \ .$$

*Proof.* From the definition of the ranking loss, we have

$$\mathbb{E}[\mathrm{R}_n] = \sum_{i=1}^{L} \sum_{j=1}^{K} \sum_{k=j+1}^{K} \mathbb{E}[\mathbb{1}\{\hat{\sigma}_{n,i}(j) > \hat{\sigma}_{n,i}(k)\}] = \sum_{i=1}^{L} \sum_{j=1}^{K} \sum_{k=j+1}^{K} \mathbb{P}\left(\mathbf{x}_{i,j}^\top \hat{\boldsymbol{\theta}}_n < \mathbf{x}_{i,k}^\top \hat{\boldsymbol{\theta}}_n\right) \ ,$$

where $\mathbb{P}\left(\mathbf{x}_{i,j}^\top \hat{\boldsymbol{\theta}}_n < \mathbf{x}_{i,k}^\top \hat{\boldsymbol{\theta}}_n\right)$ is the probability of predicting a sub-optimal item $k$ above item $j$ in list $i$. We bound this probability from above by bounding the sum of $\mathbb{P}\left(\mathbf{x}_{i,k}^\top (\hat{\boldsymbol{\theta}}_n - \boldsymbol{\theta}_*) > \frac{\Delta_{i,j,k}}{2}\right)$ and $\mathbb{P}\left(\mathbf{x}_{i,j}^\top (\boldsymbol{\theta}_* - \hat{\boldsymbol{\theta}}_n) > \frac{\Delta_{i,j,k}}{2}\right)$. Each of these probabilities is bounded from above by $\exp\left[-\frac{\Delta_{i,j,k}^2 n}{8d}\right]$, using a concentration inequality in Lemma 8. The full proof is in Appendix A.4. $\square$

Each term in Theorem 4 can be bounded from above by $\exp\left[-\frac{\Delta_{\min}^2 n}{8d}\right]$, where $n$ is the sample size, $d$ is the number of features, and $\Delta_{\min}$ denotes the minimum gap. Therefore, the bound decreases

exponentially with budget $n$ and gaps, and increases with $d$. This dependence is similar to that in Theorem 1 of Azizi et al. [6] for fixed-budget best-arm identification in linear models. Yang and Tan [95] derived a similar bound and a matching lower bound. The gaps $\Delta_{i,j,k}$ reflect the hardness of sorting list $i$, which depends on the differences of the mean rewards of items $j$ and $k$ in it.

Finally, we wanted to note that our optimal designs may not be optimal for ranking. We have not focused solely on ranking because we see value in both prediction error (Theorem 3) and ranking loss (Theorem 4) bounds. The fact that we provide both shows the versatility of our approach.

## 5 Learning with Ranking Feedback

This section is organized similarly to Section 4. In Section 5.1, we present an algorithm for human preference elicitation under ranking feedback. We bound its prediction error in Section 5.2 and its ranking loss in Section 5.3. Our algorithm design and analysis are under the following assumption, which we borrow from Zhu et al. [102].

**Assumption 1.** *We assume that the model parameter satisfies $\boldsymbol{\theta}_* \in \boldsymbol{\Theta}$, where*

$$\boldsymbol{\Theta} = \{\boldsymbol{\theta} \in \mathbb{R}^d : \boldsymbol{\theta}^\top \mathbf{1}_d = 0, \|\boldsymbol{\theta}\|_2 \le 1\}. \tag{9}$$

*We also assume that $\max_{i \in [L], k \in [K]} \|\mathbf{x}_{i,k}\|_2 \le 1$.*

The assumption of bounded model parameter and feature vectors is common in bandits [1, 45]. The additional assumption of $\boldsymbol{\theta}^\top \mathbf{1}_d = 0$ is from Zhu et al. [102], from which we borrow the estimator and concentration bound.

### 5.1 Algorithm Dope

Our algorithm for ranking feedback is similar to Dope in Section 4. It also has four main parts. First, we solve the optimal design problem in (6) to obtain a data logging policy $\pi_*$. The matrix for list $i$ is $\mathbf{A}_i = [\mathbf{z}_{i,j,k}]_{(j,k) \in \Pi_2(K)} \in \mathbb{R}^{d \times K(K-1)/2}$, where $\mathbf{z}_{i,j,k} = \mathbf{x}_{i,j} - \mathbf{x}_{i,k}$ is the difference of feature vectors of items $j$ and $k$ in list $i$. Second, we collect human feedback for $n$ rounds. At round $t \in [n]$, we sample a list $I_t \sim \pi_*$ and then observe $\sigma_t$ drawn from the PL model, as defined in (2). Third, we estimate the model parameter as

$$\hat{\boldsymbol{\theta}}_n = \arg\min_{\boldsymbol{\theta} \in \boldsymbol{\Theta}} \ell_n(\boldsymbol{\theta}), \quad \ell_n(\boldsymbol{\theta}) = -\frac{1}{n} \sum_{t=1}^n \sum_{k=1}^K \log \left( \frac{\exp[\mathbf{x}_{I_t, \sigma_t(k)}^\top \boldsymbol{\theta}]}{\sum_{j=k}^K \exp[\mathbf{x}_{I_t, \sigma_t(j)}^\top \boldsymbol{\theta}]} \right), \tag{10}$$

where $\boldsymbol{\Theta}$ is defined in Assumption 1. We solve this estimation problem using *iteratively reweighted least squares (IRLS)* [89], a popular method for fitting the parameters of *generalized linear models (GLMs)*. Finally, we sort the items in all lists $i$ according to their estimated mean rewards $\mathbf{x}_{i,k}^\top \hat{\boldsymbol{\theta}}_n$ in descending order, to obtain the permutation $\hat{\sigma}_{n,i}$. The pseudo-code of Dope is in Algorithm 2.

The optimal design for (10) is derived as follows. First, we derive the Hessian of $\ell_n(\boldsymbol{\theta})$, $\nabla^2 \ell_n(\boldsymbol{\theta})$, in Lemma 9. The optimal design with $\nabla^2 \ell_n(\boldsymbol{\theta})$ cannot be solved exactly because $\nabla^2 \ell_n(\boldsymbol{\theta})$ depends on an unknown model parameter $\boldsymbol{\theta}$. To get around this, we bound $\boldsymbol{\theta}$-dependent terms from below. Many prior works on decision making under uncertainty with GLMs [26, 52, 102, 20, 99] took a similar approach. We derive normalized and unnormalized covariance matrices

$$\boldsymbol{\Sigma}_n = \frac{2}{K(K-1)n} \bar{\boldsymbol{\Sigma}}_n, \quad \bar{\boldsymbol{\Sigma}}_n = \sum_{t=1}^n \sum_{j=1}^K \sum_{k=j+1}^K \mathbf{z}_{I_t,j,k} \mathbf{z}_{I_t,j,k}^\top, \tag{11}$$

and prove that $\nabla^2 \ell_n(\boldsymbol{\theta}) \succeq \gamma \boldsymbol{\Sigma}_n$ for some $\gamma > 0$. Therefore, we can maximize $\log \det(\nabla^2 \ell_n(\boldsymbol{\theta}))$, for any $\boldsymbol{\theta} \in \boldsymbol{\Theta}$, by maximizing $\log \det(\boldsymbol{\Sigma}_n)$. The matrix for list $i$, $\mathbf{A}_i$, can be related to the inner sum in (11) through $\operatorname{tr}(\mathbf{A}_i \mathbf{A}_i^\top) = \sum_{j=1}^K \sum_{k=j+1}^K \mathbf{z}_{i,j,k} \mathbf{z}_{i,j,k}^\top$.

The price to pay for our approximation is a constant $C > 0$ in our bounds (Theorems 5 and 6). In Appendix C, we discuss a more adaptive design and also compare to it empirically. We conclude that it would be harder to implement and analyze, and we do not observe empirical benefits at $K = 2$.

## 5.2 Maximum Prediction Error Under Ranking Feedback

In this section, we bound the maximum prediction error of Dope under ranking feedback. Similarly to the proof of Theorem 3, we decompose the error into two parts, which capture the efficiency of the optimal design and the uncertainty in the MLE $\hat{\boldsymbol{\theta}}_n$.

**Theorem 5** (Maximum prediction error). *With probability at least $1 - \delta$, the maximum prediction error after $n$ rounds is*

$$\max_{i \in [L]} \operatorname{tr}(\mathbf{A}_i^\top (\hat{\boldsymbol{\theta}}_n - \boldsymbol{\theta}_*)(\hat{\boldsymbol{\theta}}_n - \boldsymbol{\theta}_*)^\top \mathbf{A}_i) = O\left(\frac{K^6(d^2 + d\log(1/\delta))}{n}\right) .$$

This theorem is proved in Appendix A.5. We build on a self-normalizing bound of Zhu et al. [102], $\|\hat{\boldsymbol{\theta}}_n - \boldsymbol{\theta}_*\|_{\bar{\boldsymbol{\Sigma}}_n}^2 \le O\left(\frac{K^4(d + \log(1/\delta))}{n}\right)$, which may not be tight in $K$. If the bound could be improved by a multiplicative $c > 0$, we would get a multiplicative $c$ improvement in Theorem 5. Note that if the allocations $n\pi_*(i)$ are not integers, a rounding procedure is necessary [66, 25, 37]. This would result in an additional multiplicative $1 + \beta$ in our bound, for some $\beta > 0$. We omit this factor in our derivations to simplify them.

## 5.3 Ranking Loss Under Ranking Feedback

In this section, we bound the expected ranking loss under ranking feedback. Similarly to Section 4.3, we define the *gap* between the mean rewards of items $j$ and $k$ in list $i$ as $\Delta_{i,j,k} = \mathbf{z}_{i,j,k}^\top \boldsymbol{\theta}_*$, where $\mathbf{z}_{i,j,k} = \mathbf{x}_{i,j} - \mathbf{x}_{i,k}$ is the difference of feature vectors of items $j$ and $k$ in list $i$.

**Theorem 6** (Ranking loss). *The expected ranking loss after $n$ rounds is bounded as*

$$\mathbb{E}[\mathrm{R}_n] \le \sum_{i=1}^{L} \sum_{j=1}^{K} \sum_{k=j+1}^{K} \exp\left[-\frac{\Delta_{i,j,k}^2 n}{CK^4 d} + d\right] ,$$

*where $C > 0$ is a constant.*

*Proof.* The proof is similar to Theorem 4. At the end of round $n$, we bound the probability that a sub-optimal item $k$ is ranked above item $j$. The proof has two parts. First, for any list $i \in [L]$ and items $(j, k) \in \Pi_2(K)$, we show that $\mathbb{P}\left(\mathbf{x}_{i,j}^\top \hat{\boldsymbol{\theta}}_n < \mathbf{x}_{i,k}^\top \hat{\boldsymbol{\theta}}_n\right) = \mathbb{P}\left(\mathbf{z}_{i,j,k}^\top (\boldsymbol{\theta}_* - \hat{\boldsymbol{\theta}}_n) > \Delta_{i,j,k}\right)$. Then we bound this quantity by $\exp\left[-\frac{\Delta_{i,j,k}^2 n}{CK^4 d} + d\right]$. The full proof is in Appendix A.6. □

The bound in Theorem 6 is similar to that in Theorem 4, with the exception of multiplicative $K^{-4}$ and additive $d$. The leading term inside the sum can be bounded by $\exp\left[-\frac{\Delta_{\min}^2 n}{CK^4 d}\right]$, where $n$ is the sample size, $d$ is the number of features, and $\Delta_{\min}$ is the minimum gap. Therefore, similarly to Theorem 4, the bound decreases exponentially with budget $n$ and gaps, and increases with $d$. This dependence is similar to Theorem 2 of Azizi et al. [6] for fixed-budget best-arm identification in GLMs. Our bound does not involve the extra factor of $\kappa > 0$ because we assume that all vectors lie in a unit ball (Assumption 1).

## 6 Experiments

The goal of our experiments is to evaluate Dope empirically and compare it to baselines. All methods estimate $\hat{\boldsymbol{\theta}}_n$ using (7) or (10), depending on the feedback. To guarantee that these problems are well defined, even if the sample covariance matrix $\bar{\boldsymbol{\Sigma}}_n$ is not full rank, we regularize both objectives with $\gamma\|\boldsymbol{\theta}\|_2^2$, for a small $\gamma > 0$. This mostly impacts small sample sizes. Specifically, since the optimal design collects diverse feature vectors, $\bar{\boldsymbol{\Sigma}}_n$ is likely to be full rank for large sample sizes. After $\hat{\boldsymbol{\theta}}_n$ is estimated, each method ranks items in all lists based on their estimated mean rewards $\mathbf{x}_{i,k}^\top \hat{\boldsymbol{\theta}}_n$. The performance of all methods is measured by their ranking loss in (3) divided by $L$. All experiments are averaged over 100 independent runs, and we report results in Figure 1. We compare the following algorithms:

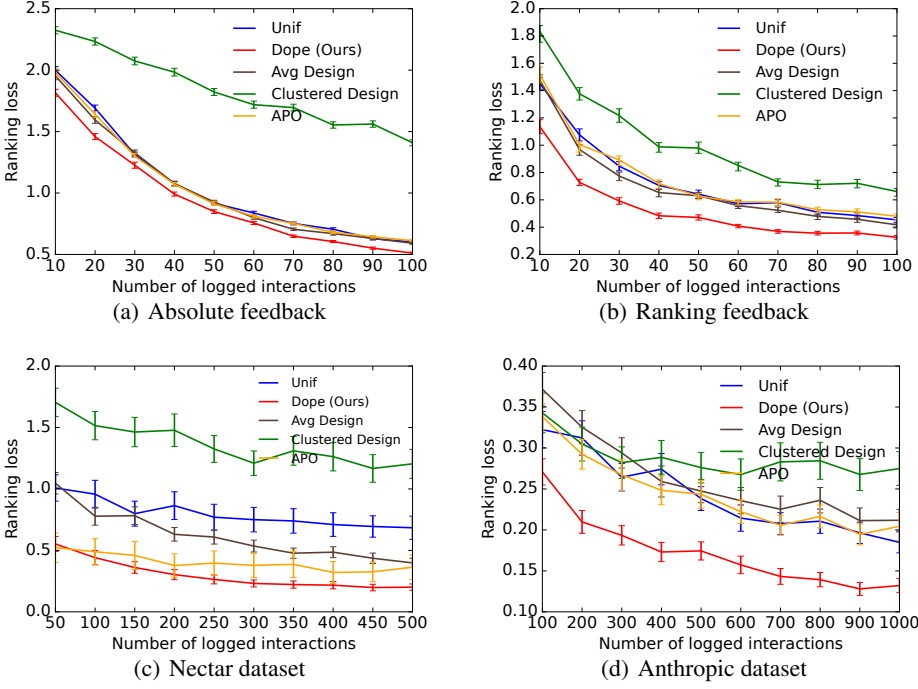

Figure 1: Ranking loss of all compared methods as a function of the number of rounds. The error bars are one standard error of the estimates.

**(1) Dope**: This is our method. We solve the optimal design problem in (6) and then sample lists $I_t$ according to $\pi_*$.

**(2) Unif**: This baseline chooses lists $I_t$ uniformly at random from $[L]$. While simple, it is known to be competitive in real-world problems where feature vectors may cover the feature space close to uniformly [4, 96, 3, 70].

**(3) Avg-Design**: The exploration policy is an optimal design over feature vectors. The feature vector of list $i$ is the mean of the feature vectors of all items in it, $\bar{\mathbf{x}}_i = \frac{1}{K} \sum_{k=1}^{K} \mathbf{x}_{i,k}$. After the design is computed, we sample lists $I_t$ according to it. The rest is the same as in Dope. This baseline shows that our list representation with multiple feature vectors can outperform more naive choices.

**(4) Clustered-Design**: This approach uses the same representation as Avg-Design. The difference is that we cluster the lists using $k$-medoids. Then we sample lists $I_t$ uniformly at random from the cluster centroids. The rest is the same as in Avg-Design. This baseline shows that Dope outperforms other notions of diversity, such as obtained by clustering. We tune $k$ ($k = 10$ in the Nectar dataset and $k = 6$ otherwise) and report only the best results.

**(5) APO**: This method was proposed in Das et al. [20] and is the closest related work. APO greedily minimizes the maximum error in pairwise ranking of $L$ lists of length $K = 2$. We extend it to $K > 2$ as follows. First, we turn $L$ lists of length $K$ into $\binom{K}{2} L$ lists of length 2, one for each pair of items in the original lists. Then we apply APO to these $\binom{K}{2} L$ lists of length 2.

Pure exploration algorithms are often compared to cumulative regret baselines [13, 5]. Since our problem is a form of learning to rank, *online learning to rank (OLTR)* baselines [67, 43, 105] seem natural. We do not compare to them for the following reason. The problem of an optimal design over lists is to design a distribution over queries. All OLTR algorithms solve a different problem, return a ranked list of items conditioned on a query chosen by the environment. Since they do not choose the queries, they cannot solve our problem.

**Synthetic experiment 1 (absolute feedback):** We have $L = 400$ questions and represent them by random vectors $\mathbf{q}_i \in [-1, 1]^6$. Each question has $K = 4$ answers. For each question, we generate $K$ random answers $\mathbf{a}_{i,k} \in [-1, 1]^6$. Both the question and answer vectors are normalized to unit length.

For each question-answer pair $(i, k)$, the feature vector is $\mathbf{x}_{i,k} = \text{vec}(\mathbf{q}_i \mathbf{a}_{i,k}^\top)$ and has length $d = 36$. The outer product captures cross-interaction terms of the question and answer representations. A similar technique has been used for feature preprocessing of the Yahoo! Front Page Today Module User Click Log Dataset [50, 51, 103, 7]. We choose a random $\boldsymbol{\theta}_* \in [0, 1]^d$. The absolute feedback is generated as in (1). Our results are reported in Figure 1(a). We note that the ranking loss of Dope decreases the fastest among all methods, with Unif, Avg-Design, and APO being close second.

**Synthetic experiment 2 (ranking feedback):** This experiment is similar to the first experiment, except that the feedback is generated by the PL model in (2). Our results are reported in Figure 1(b) and we observe again that the ranking loss of Dope decreases the fastest. The closest baselines are Unif, Avg-Design, and APO. Their lowest ranking loss ($n = 100$) is attained by Dope at $n = 60$, which is nearly a two-fold reduction in sample size. In Appendix E, we conduct additional studies on this problem. We vary the number of lists $L$ and items $K$, and report the computation time and ranking loss.

**Experiment 3 (Nectar dataset):** The Nectar dataset [101] is a dataset of 183k questions, each with 7 answers. We take a subset of this dataset: $L = 2\,000$ questions and $K = 5$ answers. The answers are generated by GPT-4, GPT-4-0613, GPT-3.5-turbo, GPT-3.5-turbo-instruct, and Anthropic models. We embed the questions and answers in 768 dimensions using Instructor embeddings [79]. Then we project them to $\mathbb{R}^{10}$ using a random projection matrix. The feature vector of answer $k$ to question $i$ is $\mathbf{x}_{i,k} = \text{vec}(\mathbf{q}_i \mathbf{a}_{i,k}^\top)$, where $\mathbf{q}_i$ and $\mathbf{a}_{i,k}$ are the projected embeddings of question $i$ and answer $k$, respectively. Hence $d = 100$. The ranking feedback is simulated using the PL model in (2). We estimate its parameter $\boldsymbol{\theta}_* \in \mathbb{R}^d$ from the ranking feedback in the dataset using the MLE in (10). Our results are reported in Figure 1(c). We observe that the ranking loss of Dope is the lowest. The closest baseline is APO. Its lowest ranking loss ($n = 500$) is attained by Dope at $n = 150$, which is more than a three-fold reduction in sample size.

**Experiment 4 (Anthropic dataset):** The Anthropic dataset [8] is a dataset of 161k questions with two answers per question. We take a subset of $L = 2\,000$ questions. We embed the questions and answers in 768 dimensions using Instructor embeddings [79]. Then we project them to $\mathbb{R}^6$ using a random projection matrix. The feature vector of answer $k$ to question $i$ is $\mathbf{x}_{i,k} = \text{vec}(\mathbf{q}_i \mathbf{a}_{i,k}^\top)$, where $\mathbf{q}_i$ and $\mathbf{a}_{i,k}$ are the projected embeddings of question $i$ and answer $k$, respectively. Hence $d = 36$. The ranking feedback is simulated using the PL model in (2). We estimate its parameter $\boldsymbol{\theta}_* \in \mathbb{R}^d$ from the feedback in the dataset using the MLE in (10). Our results are reported in Figure 1(d). We note again that the ranking loss of Dope is the lowest. The closest baselines are Unif, Avg-Design, and APO. Their lowest ranking loss ($n = 1\,000$) is attained by Dope at $n = 300$, which is more than a three-fold reduction in sample size.

## 7   Conclusions

We study the problem of optimal human preference elicitation for learning preference models. The problem is formalized as learning to rank $K$ answers to $L$ questions under a budget on the number of asked questions. We consider two feedback models: absolute and ranking. The absolute feedback is motivated by how humans assign relevance judgments in search [30, 57]. The ranking feedback is motivated by learning reward models in RLHF [39, 68, 36, 15, 77, 17]. We address both settings in a unified way. The key idea in our work is to generalize optimal designs [41, 45], a methodology for computing optimal information-gathering policies, to ranked lists. After the human feedback is collected, we learn preference models using existing estimators. Our method is statistically efficient, computationally efficient, and can be analyzed. We bound its prediction errors and ranking losses, in both absolute and ranking feedback models, and evaluate it empirically to show that it is practical.

Our work can be extended in several directions. First, we study only two models of human feedback: absolute and ranking. However, many feedback models exist [34]. One common property of these models is that learning of human preferences can be formulated as likelihood maximization. In such cases, an optimal design exists and can be used for human preference elicitation, exactly as in our work. Second, while we bound the prediction errors and ranking losses of Dope, we do not derive matching lower bounds. Therefore, although we believe that Dope is near optimal, we do not prove it. Third, we want to extend our methodology to the fixed-confidence setting. Finally, we want to apply our approach to learning reward models in LLMs and evaluate it.

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

# A Proofs

This section contains proofs of our main claims.

## A.1 Proof of Theorem 1

We follow the sketch of the proof in Section 21.1 of Lattimore and Szepesvari [45] and adapt it to matrices. Before we start, we prove several helpful claims.

First, using (43) in Petersen and Pedersen [64], we have

$$\frac{\partial}{\partial \pi(i)} f(\pi) = \frac{\partial}{\partial \pi(i)} \log \det(\mathbf{V}_\pi) = \operatorname{tr}\left(\mathbf{V}_\pi^{-1} \frac{\partial}{\partial \pi(i)} \mathbf{V}_\pi\right) = \operatorname{tr}(\mathbf{V}_\pi^{-1} \mathbf{A}_i \mathbf{A}_i^\top) = \operatorname{tr}(\mathbf{A}_i^\top \mathbf{V}_\pi^{-1} \mathbf{A}_i).$$

In the last step, we use the cyclic property of the trace. We define the gradient of $f(\pi)$ with respect to $\pi$ as $\nabla f(\pi) = (\operatorname{tr}(\mathbf{A}_i^\top \mathbf{V}_\pi^{-1} \mathbf{A}_i))_{i=1}^L$. Second, using basic properties of the trace, we have

$$\sum_{i=1}^L \pi(i) \operatorname{tr}(\mathbf{A}_i^\top \mathbf{V}_\pi^{-1} \mathbf{A}_i) = \sum_{i=1}^L \pi(i) \operatorname{tr}(\mathbf{V}_\pi^{-1} \mathbf{A}_i \mathbf{A}_i^\top) = \operatorname{tr}\left(\sum_{i=1}^L \pi(i) \mathbf{V}_\pi^{-1} \mathbf{A}_i \mathbf{A}_i^\top\right) \tag{12}$$

$$= \operatorname{tr}\left(\mathbf{V}_\pi^{-1} \sum_{i=1}^L \pi(i) \mathbf{A}_i \mathbf{A}_i^\top\right) = \operatorname{tr}(I_d) = d.$$

Finally, for any distribution $\pi$, (12) implies

$$g(\pi) = \max_{i \in [L]} \operatorname{tr}(\mathbf{A}_i^\top \mathbf{V}_\pi^{-1} \mathbf{A}_i) \geq \sum_{i=1}^L \pi(i) \operatorname{tr}(\mathbf{A}_i^\top \mathbf{V}_\pi^{-1} \mathbf{A}_i) = d. \tag{13}$$

Now we are ready to start the proof.

$(b) \Rightarrow (a)$: Let $\pi_*$ be a maximizer of $f(\pi)$. By first-order optimality conditions, for any distribution $\pi$, we have

$$0 \geq \langle \nabla f(\pi_*), \pi - \pi_* \rangle = \sum_{i=1}^L \pi(i) \operatorname{tr}(\mathbf{A}_i^\top \mathbf{V}_{\pi_*}^{-1} \mathbf{A}_i) - \sum_{i=1}^L \pi_*(i) \operatorname{tr}(\mathbf{A}_i^\top \mathbf{V}_{\pi_*}^{-1} \mathbf{A}_i)$$

$$= \sum_{i=1}^L \pi(i) \operatorname{tr}(\mathbf{A}_i^\top \mathbf{V}_{\pi_*}^{-1} \mathbf{A}_i) - d.$$

In the last step, we use (12). Since this inequality holds for any distribution $\pi$, including Dirac at $i$ for any $i \in [L]$, we have $d \geq \max_{i \in [L]} \operatorname{tr}(\mathbf{A}_i^\top \mathbf{V}_{\pi_*}^{-1} \mathbf{A}_i) = g(\pi_*)$. Finally, by (13), $g(\pi) \geq d$ holds for any distribution $\pi$. Therefore, $\pi_*$ must be a minimizer of $g(\pi)$ and $g(\pi_*) = d$.

$(c) \Rightarrow (b)$: Note that

$$\langle \nabla f(\pi_*), \pi - \pi_* \rangle = \sum_{i=1}^L \pi(i) \operatorname{tr}(\mathbf{A}_i^\top \mathbf{V}_{\pi_*}^{-1} \mathbf{A}_i) - d \leq \max_{i \in [L]} \operatorname{tr}(\mathbf{A}_i^\top \mathbf{V}_{\pi_*}^{-1} \mathbf{A}_i) - d = g(\pi_*) - d$$

holds for any distributions $\pi$ and $\pi_*$. Since $g(\pi_*) = d$, we have $\langle \nabla f(\pi_*), \pi - \pi_* \rangle \leq 0$. Therefore, by first-order optimality conditions, $\pi_*$ is a maximizer of $f(\pi)$.

$(a) \Rightarrow (c)$: This follows from the same argument as in $(b) \Rightarrow (a)$. In particular, any maximizer $\pi_*$ of $f(\pi)$ is a minimizer of $g(\pi)$, and $g(\pi_*) = d$.

To prove that $|\operatorname{supp}(\pi_*)| \leq d(d+1)/2$, we argue that $\pi_*$ can be substituted for a distribution with a lower support whenever $|\operatorname{supp}(\pi_*)| > d(d+1)/2$. The claim then follows by induction.

Let $S = \operatorname{supp}(\pi_*)$ and suppose that $|S| > d(d+1)/2$. We start with designing a suitable family of optimal solutions. Since the space of $d \times d$ symmetric matrices has $d(d+1)/2$ dimensions, there must exist an $L$-dimensional vector $\eta$ such that $\operatorname{supp}(\eta) \subseteq S$ and

$$\sum_{i \in S} \eta(i) \mathbf{A}_i \mathbf{A}_i^\top = \mathbf{0}_{d,d}, \tag{14}$$

where $\mathbf{0}_{d,d}$ is a $d \times d$ zero matrix. Let $\pi_t = \pi_* + t\eta$ for $t \geq 0$. An important property of $\pi_t$ is that

$$\log \det(\mathbf{V}_{\pi_t}) = \log \det \left( \mathbf{V}_{\pi_*} + t \sum_{i \in S} \eta(i) \mathbf{A}_i \mathbf{A}_i^\top \right) = \log \det(\mathbf{V}_{\pi_*}).$$

Therefore, any $\pi_t$ is an optimal solution. However, it may not be a distribution.

We prove that $\pi_t \in \Delta^L$, for some $t > 0$, as follows. First, note that $\mathrm{tr}(\mathbf{A}_i^\top \mathbf{V}_{\pi_*}^{-1} \mathbf{A}_i) = d$ holds for all $i \in S$. Otherwise $\pi_*$ could be improved. Using this observation, we have

$$d \sum_{i \in S} \eta(i) = \sum_{i \in S} \eta(i) \, \mathrm{tr}(\mathbf{A}_i^\top \mathbf{V}_{\pi_*}^{-1} \mathbf{A}_i) = \sum_{i \in S} \eta(i) \, \mathrm{tr}(\mathbf{V}_{\pi_*}^{-1} \mathbf{A}_i \mathbf{A}_i^\top)$$

$$= \mathrm{tr} \left( \mathbf{V}_{\pi_*}^{-1} \sum_{i \in S} \eta(i) \mathbf{A}_i \mathbf{A}_i^\top \right) = 0 \,,$$

where the last equality follows from (14). This implies that $\sum_{i \in S} \eta(i) = 0$ and that $\pi_t \in \Delta^L$, for as long as $\pi_t \geq \mathbf{0}_L$.

Finally, we take the largest feasible $t$, $\tau = \max\{t > 0 : \pi_t \in \Delta^L\}$, and note that $\pi_\tau$ has at least one more non-zero entry than $\pi_*$ while having the same value. This concludes the proof.

### A.2   Proof of Lemma 2

We note that for any list $i \in [L]$,

$$\sum_{\mathbf{a} \in \mathbf{A}_i} \|\mathbf{a}\|_{\bar{\mathbf{\Sigma}}_n^{-1}}^2 = \mathrm{tr}(\mathbf{A}_i^\top \bar{\mathbf{\Sigma}}_n^{-1} \mathbf{A}_i) = \mathrm{tr} \left( \mathbf{A}_i^\top \left( \sum_{t=1}^n \sum_{k=1}^K \mathbf{x}_{I_t,k} \mathbf{x}_{I_t,k}^\top \right)^{-1} \mathbf{A}_i \right)$$

$$= \frac{1}{n} \mathrm{tr} \left( \mathbf{A}_i^\top \left( \sum_{i=1}^L \pi_*(i) \sum_{k=1}^K \mathbf{x}_{i,k} \mathbf{x}_{i,k}^\top \right)^{-1} \mathbf{A}_i \right) = \frac{1}{n} \mathrm{tr}(\mathbf{A}_i^\top \mathbf{V}_{\pi_*}^{-1} \mathbf{A}_i) \,.$$

The third equality holds because all $n\pi_*(i)$ are integers and $n > 0$. In this case, the optimal design is exact and $\bar{\mathbf{\Sigma}}_n$ invertible, because all of its eigenvalues are positive. Now we use the definition of $g(\pi_*)$, apply Theorem 1, and get that

$$\max_{i \in [L]} \mathrm{tr}(\mathbf{A}_i^\top \mathbf{V}_{\pi_*}^{-1} \mathbf{A}_i) = g(\pi_*) = d \,.$$

This concludes the proof.

### A.3   Proof of Theorem 3

For any list $i \in [L]$, we have

$$\mathrm{tr}(\mathbf{A}_i^\top (\hat{\boldsymbol{\theta}}_n - \boldsymbol{\theta}_*)(\hat{\boldsymbol{\theta}}_n - \boldsymbol{\theta}_*)^\top \mathbf{A}_i) = \sum_{\mathbf{a} \in \mathbf{A}_i} (\mathbf{a}^\top (\hat{\boldsymbol{\theta}}_n - \boldsymbol{\theta}_*))^2 = \sum_{\mathbf{a} \in \mathbf{A}_i} (\mathbf{a}^\top \bar{\mathbf{\Sigma}}_n^{-1/2} \bar{\mathbf{\Sigma}}_n^{1/2} (\hat{\boldsymbol{\theta}}_n - \boldsymbol{\theta}_*))^2$$

$$\leq \sum_{\mathbf{a} \in \mathbf{A}_i} \|\mathbf{a}\|_{\bar{\mathbf{\Sigma}}_n^{-1}}^2 \|\hat{\boldsymbol{\theta}}_n - \boldsymbol{\theta}_*\|_{\bar{\mathbf{\Sigma}}_n}^2 \,,$$

where the last step follows from the Cauchy-Schwarz inequality. Therefore,

$$\max_{i \in [L]} \mathrm{tr}(\mathbf{A}_i^\top (\hat{\boldsymbol{\theta}}_n - \boldsymbol{\theta}_*)(\hat{\boldsymbol{\theta}}_n - \boldsymbol{\theta}_*)^\top \mathbf{A}_i) \leq \max_{i \in [L]} \sum_{\mathbf{a} \in \mathbf{A}_i} \|\mathbf{a}\|_{\bar{\mathbf{\Sigma}}_n^{-1}}^2 \|\hat{\boldsymbol{\theta}}_n - \boldsymbol{\theta}_*\|_{\bar{\mathbf{\Sigma}}_n}^2$$

$$= \max_{i \in [L]} \underbrace{\sum_{\mathbf{a} \in \mathbf{A}_i} \|\mathbf{a}\|_{\bar{\mathbf{\Sigma}}_n^{-1}}^2}_{\text{Part I}} \underbrace{n\|\hat{\boldsymbol{\theta}}_n - \boldsymbol{\theta}_*\|_{\mathbf{\Sigma}_n}^2}_{\text{Part II}} \,,$$

where we use $\bar{\mathbf{\Sigma}}_n = n\mathbf{\Sigma}_n$ in the last step.

Part I captures the efficiency of data collection and depends on the optimal design. By Lemma 2,

$$\max_{i \in [L]} \sum_{\mathbf{a} \in \mathbf{A}_i} \|\mathbf{a}\|^2_{\bar{\mathbf{\Sigma}}_n^{-1}} = \frac{d}{n} \,.$$

Part II measures how the estimated model parameter $\hat{\boldsymbol{\theta}}_n$ differs from the true model parameter $\boldsymbol{\theta}_*$, under the empirical covariance matrix $\mathbf{\Sigma}_n$. To bound this term, we use Lemma 7 and get that

$$\|\hat{\boldsymbol{\theta}}_n - \boldsymbol{\theta}_*\|^2_{\mathbf{\Sigma}_n} \leq \frac{16d + 8\log(1/\delta)}{n}$$

holds with probability at least $1 - \delta$. The main claim follows from combining the upper bounds on Parts I and II.

### A.4  Proof of Theorem 4

From the definition of ranking loss, we have

$$\mathbb{E}[\mathrm{R}_n] = \sum_{i=1}^{L} \sum_{j=1}^{K} \sum_{k=j+1}^{K} \mathbb{E}[\mathbb{1}\{\hat{\sigma}_{n,i}(j) > \hat{\sigma}_{n,i}(k)\}] = \sum_{i=1}^{L} \sum_{j=1}^{K} \sum_{k=j+1}^{K} \mathbb{P}\left(\mathbf{x}_{i,j}^\top \hat{\boldsymbol{\theta}}_n < \mathbf{x}_{i,k}^\top \hat{\boldsymbol{\theta}}_n\right) \,.$$

In the rest of the proof, we bound each term separately. Specifically, for any list $i \in [L]$ and items $(j,k) \in \Pi_2(K)$ in it, we have

$$\begin{aligned}
\mathbb{P}\left(\mathbf{x}_{i,j}^\top \hat{\boldsymbol{\theta}}_n < \mathbf{x}_{i,k}^\top \hat{\boldsymbol{\theta}}_n\right) &= \mathbb{P}\left(\mathbf{x}_{i,k}^\top \hat{\boldsymbol{\theta}}_n - \mathbf{x}_{i,j}^\top \hat{\boldsymbol{\theta}}_n > 0\right) \\
&= \mathbb{P}\left(\mathbf{x}_{i,k}^\top \hat{\boldsymbol{\theta}}_n - \mathbf{x}_{i,j}^\top \hat{\boldsymbol{\theta}}_n + \Delta_{i,j,k} > \Delta_{i,j,k}\right) \\
&= \mathbb{P}\left(\mathbf{x}_{i,k}^\top \hat{\boldsymbol{\theta}}_n - \mathbf{x}_{i,j}^\top \hat{\boldsymbol{\theta}}_n + \mathbf{x}_{i,j}^\top \boldsymbol{\theta}_* - \mathbf{x}_{i,k}^\top \boldsymbol{\theta}_* > \Delta_{i,j,k}\right) \\
&= \mathbb{P}\left(\mathbf{x}_{i,k}^\top (\hat{\boldsymbol{\theta}}_n - \boldsymbol{\theta}_*) + \mathbf{x}_{i,j}^\top (\boldsymbol{\theta}_* - \hat{\boldsymbol{\theta}}_n) > \Delta_{i,j,k}\right) \\
&\leq \mathbb{P}\left(\mathbf{x}_{i,k}^\top (\hat{\boldsymbol{\theta}}_n - \boldsymbol{\theta}_*) > \frac{\Delta_{i,j,k}}{2}\right) + \mathbb{P}\left(\mathbf{x}_{i,j}^\top (\boldsymbol{\theta}_* - \hat{\boldsymbol{\theta}}_n) > \frac{\Delta_{i,j,k}}{2}\right) \,.
\end{aligned}$$

In the third equality, we use that $\Delta_{i,j,k} = (\mathbf{x}_{i,j} - \mathbf{x}_{i,k})^\top \boldsymbol{\theta}_*$. The last step follows from the fact that event $A + B > c$ occurs only if $A > c/2$ or $B > c/2$.

Now we bound $\mathbb{P}\left(\mathbf{x}_{i,k}^\top (\hat{\boldsymbol{\theta}}_n - \boldsymbol{\theta}_*) > \Delta_{i,j,k}/2\right)$ and note that the other term can be bounded analogously. Specifically, we apply Lemmas 2 and 8, and get

$$\mathbb{P}\left(\mathbf{x}_{i,k}^\top (\hat{\boldsymbol{\theta}}_n - \boldsymbol{\theta}_*) > \frac{\Delta_{i,j,k}}{2}\right) \leq \exp\left[-\frac{\Delta_{i,j,k}^2}{8\|\mathbf{x}_{i,k}\|^2_{\bar{\mathbf{\Sigma}}_n^{-1}}}\right] \leq \exp\left[-\frac{\Delta_{i,j,k}^2 n}{8d}\right] \,.$$

This concludes the proof.

The above approach relies on the concentration of $\mathbf{x}_{i,k}^\top (\hat{\boldsymbol{\theta}}_n - \boldsymbol{\theta}_*)$, which is proved in Lemma 8. A similar result can be obtained using the Cauchy-Schwarz inequality. This is especially useful when a high-probability bound on $\|\hat{\boldsymbol{\theta}}_n - \boldsymbol{\theta}_*\|^2_{\mathbf{\Sigma}_n}$ already exists, such as in Appendix A.6. Specifically, by the Cauchy-Schwarz inequality,

$$\begin{aligned}
\mathbb{P}\left(\mathbf{x}_{i,k}^\top (\hat{\boldsymbol{\theta}}_n - \boldsymbol{\theta}_*) > \frac{\Delta_{i,j,k}}{2}\right) &\leq \mathbb{P}\left(\|\mathbf{x}_{i,k}\|_{\mathbf{\Sigma}_n^{-1}} \|\hat{\boldsymbol{\theta}}_n - \boldsymbol{\theta}_*\|_{\mathbf{\Sigma}_n} > \frac{\Delta_{i,j,k}}{2}\right) \\
&= \mathbb{P}\left(\|\mathbf{x}_{i,k}\|^2_{\mathbf{\Sigma}_n^{-1}} \|\hat{\boldsymbol{\theta}}_n - \boldsymbol{\theta}_*\|^2_{\mathbf{\Sigma}_n} > \frac{\Delta_{i,j,k}^2}{4}\right) \\
&\leq \mathbb{P}\left(\|\hat{\boldsymbol{\theta}}_n - \boldsymbol{\theta}_*\|^2_{\mathbf{\Sigma}_n} > \frac{\Delta_{i,j,k}^2}{4d}\right) \,.
\end{aligned}$$

In the second inequality, we use that $\|\mathbf{x}_{i,k}\|^2_{\mathbf{\Sigma}_n^{-1}} = n\|\mathbf{x}_{i,k}\|^2_{\hat{\mathbf{\Sigma}}_n^{-1}} \le d$, which follows from Lemma 2. Finally, Lemma 7 says that

$$\mathbb{P}\left(\|\hat{\boldsymbol{\theta}}_n - \boldsymbol{\theta}_*\|^2_{\mathbf{\Sigma}_n} \ge \frac{16d + 8\log(1/\delta)}{n}\right) \le \delta$$

holds for any $\delta > 0$. To apply this bound, we let

$$\frac{16d + 8\log(1/\delta)}{n} = \frac{\Delta_{i,j,k}^2}{4d}$$

and express $\delta$. This leads to

$$\mathbb{P}\left(\|\hat{\boldsymbol{\theta}}_n - \boldsymbol{\theta}_*\|^2_{\mathbf{\Sigma}_n} > \frac{\Delta_{i,j,k}^2}{4d}\right) \le \delta = \exp\left[-\frac{\Delta_{i,j,k}^2 n}{32d} + 2d\right],$$

which concludes the alternative proof.

## A.5 Proof of Theorem 5

Following the same steps as in Appendix A.3, we have

$$\max_{i\in[L]}\operatorname{tr}(\mathbf{A}_i^\top(\hat{\boldsymbol{\theta}}_n - \boldsymbol{\theta}_*)(\hat{\boldsymbol{\theta}}_n - \boldsymbol{\theta}_*)^\top\mathbf{A}_i) \le \underbrace{\max_{i\in[L]}\sum_{\mathbf{a}\in\mathbf{A}_i}\|\mathbf{a}\|^2_{\hat{\mathbf{\Sigma}}_n^{-1}}}_{\text{Part I}}\underbrace{\frac{K(K-1)n}{2}\|\hat{\boldsymbol{\theta}}_n - \boldsymbol{\theta}_*\|^2_{\mathbf{\Sigma}_n}}_{\text{Part II}}.$$

Part I captures the efficiency of data collection and depends on the optimal design. By Lemma 2,

$$\max_{i\in[L]}\sum_{\mathbf{a}\in\mathbf{A}_i}\|\mathbf{a}\|^2_{\hat{\mathbf{\Sigma}}_n^{-1}} = \frac{d}{n}.$$

Part II measures how the estimated model parameter $\hat{\boldsymbol{\theta}}_n$ differs from the true model parameter $\boldsymbol{\theta}_*$, under the empirical covariance matrix $\mathbf{\Sigma}_n$. To bound this term, we use Lemma 9 (a restatement of Theorem 4.1 in Zhu et al. [102]) and get that

$$\|\hat{\boldsymbol{\theta}}_n - \boldsymbol{\theta}_*\|^2_{\mathbf{\Sigma}_n} \le \frac{CK^4(d + \log(1/\delta))}{n}$$

holds with probability at least $1 - \delta$, where $C > 0$ is some constant. The main claim follows from combining the upper bounds on Parts I and II.

## A.6 Proof of Theorem 6

Following the same steps as in Appendix A.4, we get

$$
\begin{aligned}
\mathbb{P}\left(\mathbf{x}_{i,j}^\top\hat{\boldsymbol{\theta}}_n < \mathbf{x}_{i,k}^\top\hat{\boldsymbol{\theta}}_n\right) &= \mathbb{P}\left(\mathbf{x}_{i,k}^\top(\hat{\boldsymbol{\theta}}_n - \boldsymbol{\theta}_*) + \mathbf{x}_{i,j}^\top(\boldsymbol{\theta}_* - \hat{\boldsymbol{\theta}}_n) > \Delta_{i,j,k}\right) \\
&= \mathbb{P}\left(\mathbf{z}_{i,j,k}^\top(\boldsymbol{\theta}_* - \hat{\boldsymbol{\theta}}_n) > \Delta_{i,j,k}\right) \\
&\le \mathbb{P}\left(\|\mathbf{z}_{i,j,k}\|_{\mathbf{\Sigma}_n^{-1}}\|\hat{\boldsymbol{\theta}}_n - \boldsymbol{\theta}_*\|_{\mathbf{\Sigma}_n} > \Delta_{i,j,k}\right) \\
&= \mathbb{P}\left(\|\mathbf{z}_{i,j,k}\|^2_{\mathbf{\Sigma}_n^{-1}}\|\hat{\boldsymbol{\theta}}_n - \boldsymbol{\theta}_*\|^2_{\mathbf{\Sigma}_n} > \Delta_{i,j,k}^2\right) \\
&\le \mathbb{P}\left(\|\hat{\boldsymbol{\theta}}_n - \boldsymbol{\theta}_*\|^2_{\mathbf{\Sigma}_n} > \frac{\Delta_{i,j,k}^2}{d}\right).
\end{aligned}
$$

In the first inequality, we use the Cauchy-Schwarz inequality. In the second inequality, we use that $\|\mathbf{z}_{i,j,k}\|^2_{\mathbf{\Sigma}_n^{-1}} = n\|\mathbf{z}_{i,j,k}\|^2_{\hat{\mathbf{\Sigma}}_n^{-1}} \le d$, which follows from Lemma 2. Finally, Lemma 9 says that

$$\mathbb{P}\left(\|\hat{\boldsymbol{\theta}}_n - \boldsymbol{\theta}_*\|^2_{\mathbf{\Sigma}_n} \ge \frac{CK^4(d + \log(1/\delta))}{n}\right) \le \delta$$

holds for any $\delta > 0$. To apply this bound, we let

$$\frac{CK^4(d + \log(1/\delta))}{n} = \frac{\Delta_{i,j,k}^2}{d}$$

and express $\delta$. This leads to

$$\mathbb{P}\left(\|\hat{\boldsymbol{\theta}}_n - \boldsymbol{\theta}_*\|_{\boldsymbol{\Sigma}_n}^2 > \frac{\Delta_{i,j,k}^2}{d}\right) \leq \delta = \exp\left[-\frac{\Delta_{i,j,k}^2 n}{CK^4 d} + d\right],$$

which concludes the proof.

# B   Supporting Lemmas

This section contains our supporting lemmas and their proofs.

**Lemma 7.** *Consider the absolute feedback model in Section [4]. Fix $\delta \in (0, 1)$. Then*

$$\|\hat{\boldsymbol{\theta}}_n - \boldsymbol{\theta}_*\|_{\boldsymbol{\Sigma}_n}^2 \leq \frac{16d + 8\log(1/\delta)}{n}$$

*holds with probability at least $1 - \delta$.*

*Proof.* Let $\mathbf{X} \in \mathbb{R}^{Kn \times d}$ be a matrix of $Kn$ feature vectors in [(7)] and $\mathbf{y} \in \mathbb{R}^{Kn}$ be a vector of the corresponding $Kn$ observations. Under 1-sub-Gaussian noise in [(1)], we can rewrite $\hat{\boldsymbol{\theta}}_n - \boldsymbol{\theta}_*$ as

$$\hat{\boldsymbol{\theta}}_n - \boldsymbol{\theta}_* = (\mathbf{X}^\top \mathbf{X})^{-1} \mathbf{X}^\top (\mathbf{y} - \mathbf{X}\boldsymbol{\theta}_*) = (\mathbf{X}^\top \mathbf{X})^{-1} \mathbf{X}^\top \eta,$$

where $\eta \in \mathbb{R}^{Kn}$ is a vector of independent 1-sub-Gaussian noise. Now note that $\mathbf{a}^\top (\mathbf{X}^\top \mathbf{X})^{-1} \mathbf{X}^\top$ is a fixed vector of length $Kn$ for any fixed $\mathbf{a} \in \mathbb{R}^d$. Therefore, $\mathbf{a}^\top (\hat{\boldsymbol{\theta}}_n - \boldsymbol{\theta}_*)$ is a sub-Gaussian random variable with a variance proxy

$$\mathbf{a}^\top (\mathbf{X}^\top \mathbf{X})^{-1} \mathbf{X}^\top \mathbf{X} (\mathbf{X}^\top \mathbf{X})^{-1} \mathbf{a} = \mathbf{a}^\top (\mathbf{X}^\top \mathbf{X})^{-1} \mathbf{a} = \|\mathbf{a}\|_{(\mathbf{X}^\top \mathbf{X})^{-1}}^2 = \|\mathbf{a}\|_{\boldsymbol{\Sigma}_n^{-1}}^2 / n.$$

From standard concentration inequalities for sub-Gaussian random variables [11],

$$\mathbb{P}\left(\mathbf{a}^\top (\hat{\boldsymbol{\theta}}_n - \boldsymbol{\theta}_*) \geq \sqrt{\frac{2\|\mathbf{a}\|_{\boldsymbol{\Sigma}_n^{-1}}^2 \log(1/\delta)}{n}}\right) \leq \delta \tag{15}$$

holds for any fixed $\mathbf{a} \in \mathbb{R}^d$.

To bound $\|\hat{\boldsymbol{\theta}}_n - \boldsymbol{\theta}_*\|_{\boldsymbol{\Sigma}_n}$, we take advantage of the fact that

$$\|\hat{\boldsymbol{\theta}}_n - \boldsymbol{\theta}_*\|_{\boldsymbol{\Sigma}_n} = \langle \hat{\boldsymbol{\theta}}_n - \boldsymbol{\theta}_*, \boldsymbol{\Sigma}_n^{1/2} A \rangle, \quad A = \frac{\boldsymbol{\Sigma}_n^{1/2}(\hat{\boldsymbol{\theta}}_n - \boldsymbol{\theta}_*)}{\|\hat{\boldsymbol{\theta}}_n - \boldsymbol{\theta}_*\|_{\boldsymbol{\Sigma}_n}}. \tag{16}$$

While $A \in \mathbb{R}^d$ is random, it has two important properties. First, its length is 1. Second, if it was fixed, we could apply [(15)] and would get

$$\mathbb{P}\left(\langle \hat{\boldsymbol{\theta}}_n - \boldsymbol{\theta}_*, \boldsymbol{\Sigma}_n^{1/2} A \rangle \geq \sqrt{\frac{2\log(1/\delta)}{n}}\right) \leq \delta.$$

To eliminate the randomness in $A$, we use a coverage argument.

Let $\mathcal{S} = \{\mathbf{a} \in \mathbb{R}^d : \|a\|_2 = 1\}$ be a unit sphere. Lemma 20.1 in Lattimore and Szepesvari [45] says that there exists an $\varepsilon$-cover $\mathcal{C}_\varepsilon \subset \mathbb{R}^d$ of $\mathcal{S}$ that has at most $|\mathcal{C}_\varepsilon| \leq (3/\varepsilon)^d$ points. Specifically, for any $\mathbf{a} \in \mathcal{S}$, there exists $\mathbf{y} \in \mathcal{C}_\varepsilon$ such that $\|\mathbf{a} - \mathbf{y}\|_2 \leq \varepsilon$. By a union bound applied to all points in $\mathcal{C}_\varepsilon$, we have that

$$\mathbb{P}\left(\exists \mathbf{y} \in \mathcal{C}_\varepsilon : \langle \hat{\boldsymbol{\theta}}_n - \boldsymbol{\theta}_*, \boldsymbol{\Sigma}_n^{1/2} \mathbf{y} \rangle \geq \sqrt{\frac{2\log(|\mathcal{C}_\varepsilon|/\delta)}{n}}\right) \leq \delta. \tag{17}$$

Now we are ready to complete the proof. Specifically, note that

$$\|\hat{\boldsymbol{\theta}}_n - \boldsymbol{\theta}_*\|_{\boldsymbol{\Sigma}_n} \overset{(a)}{=} \max_{\mathbf{a} \in \mathcal{S}} \langle \hat{\boldsymbol{\theta}}_n - \boldsymbol{\theta}_*, \boldsymbol{\Sigma}_n^{1/2} \mathbf{a} \rangle$$

$$= \max_{\mathbf{a} \in \mathcal{S}} \min_{\mathbf{y} \in \mathcal{C}_\varepsilon} \left[ \langle \hat{\boldsymbol{\theta}}_n - \boldsymbol{\theta}_*, \boldsymbol{\Sigma}_n^{1/2}(\mathbf{a} - \mathbf{y}) \rangle + \langle \hat{\boldsymbol{\theta}}_n - \boldsymbol{\theta}_*, \boldsymbol{\Sigma}_n^{1/2} \mathbf{y} \rangle \right]$$

$$\overset{(b)}{\leq} \max_{\mathbf{a} \in \mathcal{S}} \min_{\mathbf{y} \in \mathcal{C}_\varepsilon} \left[ \|\hat{\boldsymbol{\theta}}_n - \boldsymbol{\theta}_*\|_{\boldsymbol{\Sigma}_n} \|\mathbf{a} - \mathbf{y}\|_2 + \sqrt{\frac{2 \log(|\mathcal{C}_\varepsilon|/\delta)}{n}} \right]$$

$$\overset{(c)}{\leq} \varepsilon \|\hat{\boldsymbol{\theta}}_n - \boldsymbol{\theta}_*\|_{\boldsymbol{\Sigma}_n} + \sqrt{\frac{2 \log(|\mathcal{C}_\varepsilon|/\delta)}{n}}$$

holds with probability at least $1 - \delta$. In this derivation, (a) follows from (16), (b) follows from the Cauchy-Schwarz inequality and (17), and (c) follows from the definition of $\varepsilon$-cover $\mathcal{C}_\varepsilon$. Finally, we rearrange the terms, choose $\varepsilon = 1/2$, and get that

$$\|\hat{\boldsymbol{\theta}}_n - \boldsymbol{\theta}_*\|_{\boldsymbol{\Sigma}_n} \leq 2\sqrt{\frac{2 \log(|\mathcal{C}_\varepsilon|/\delta)}{n}} \leq 2\sqrt{\frac{(2 \log 6)d + 2 \log(1/\delta)}{n}} \,.$$

This concludes the proof. □

**Lemma 8.** *Consider the absolute feedback model in Section 4. Fix $\mathbf{x} \in \mathbb{R}^d$ and $\Delta > 0$. Then*

$$\mathbb{P}\left(\mathbf{x}^\top (\hat{\boldsymbol{\theta}}_n - \boldsymbol{\theta}_*) > \Delta\right) \leq \exp\left[-\frac{\Delta^2}{2\|\mathbf{x}\|_{\hat{\boldsymbol{\Sigma}}_n^{-1}}^2}\right] \,.$$

*Proof.* The proof is from Section 2.2 in Jamieson and Jain [33]. Let $\mathbf{X} \in \mathbb{R}^{Kn \times d}$ be a matrix of $Kn$ feature vectors in (7) and $\mathbf{y} \in \mathbb{R}^{Kn}$ be a vector of the corresponding $Kn$ observations. Under 1-sub-Gaussian noise in (1), we can rewrite $\mathbf{x}^\top (\hat{\boldsymbol{\theta}}_n - \boldsymbol{\theta}_*)$ as

$$\mathbf{x}^\top (\hat{\boldsymbol{\theta}}_n - \boldsymbol{\theta}_*) = \underbrace{\mathbf{x}^\top (\mathbf{X}^\top \mathbf{X})^{-1} \mathbf{X}^\top}_{\mathbf{w}} \eta = \mathbf{w}^\top \eta = \sum_{t=1}^{Kn} \mathbf{w}_t \eta_t \,,$$

where $\mathbf{w} \in \mathbb{R}^{Kn}$ is a fixed vector and $\eta \in \mathbb{R}^{Kn}$ is a vector of independent sub-Gaussian noise. Then, for any $\Delta > 0$ and $\lambda > 0$, we have

$$\mathbb{P}\left(\mathbf{x}^\top (\hat{\boldsymbol{\theta}}_n - \boldsymbol{\theta}_*) > \Delta\right) = \mathbb{P}\left(\mathbf{w}^\top \eta > \Delta\right) = \mathbb{P}\left(\exp[\lambda \mathbf{w}^\top \eta] > \exp[\lambda \Delta]\right)$$

$$\overset{(a)}{\leq} \exp[-\lambda \Delta] \mathbb{E}\left[\exp\left[\lambda \sum_{t=1}^{Kn} \mathbf{w}_t \eta_t\right]\right] \overset{(b)}{\leq} \exp[-\lambda \Delta] \prod_{t=1}^{Kn} \mathbb{E}[\exp[\lambda \mathbf{w}_t \eta_t]]$$

$$\overset{(c)}{\leq} \exp[-\lambda \Delta] \prod_{t=1}^{Kn} \exp[\lambda^2 \mathbf{w}_t^2/2] = \exp[-\lambda \Delta + \lambda^2 \|\mathbf{w}\|_2^2/2]$$

$$\overset{(d)}{\leq} \exp\left[-\frac{\Delta^2}{2\|\mathbf{w}\|_2^2}\right] \overset{(e)}{=} \exp\left[-\frac{\Delta^2}{2\mathbf{x}^\top (\mathbf{X}^\top \mathbf{X})^{-1} \mathbf{x}}\right]$$

$$= \exp\left[-\frac{\Delta^2}{2\|\mathbf{x}\|_{\hat{\boldsymbol{\Sigma}}_n^{-1}}^2}\right] \,.$$

In this derivation, (a) follows from Markov's inequality, (b) is due to independent noise, (c) follows from sub-Gaussianity, (d) is due to setting $\lambda = \Delta/\|\mathbf{w}\|_2^2$, and (e) follows from

$$\|\mathbf{w}\|_2^2 = \mathbf{x}^\top (\mathbf{X}^\top \mathbf{X})^{-1} \mathbf{X}^\top \mathbf{X} (\mathbf{X}^\top \mathbf{X})^{-1} \mathbf{x} = \mathbf{x}^\top (\mathbf{X}^\top \mathbf{X})^{-1} \mathbf{x} \,.$$

This concludes the proof. □

**Lemma 9.** *Consider the ranking feedback model in Section 5. Fix $\delta \in (0, 1)$. Then there exists a constant $C > 0$ such that*

$$\|\hat{\boldsymbol{\theta}}_n - \boldsymbol{\theta}_*\|_{\hat{\boldsymbol{\Sigma}}_n}^2 \leq \frac{CK^4(d + \log(1/\delta))}{n}$$

*holds with probability at least $1 - \delta$.*

*Proof.* The proof has two main steps.

**Step 1:** We first prove that $\ell_n(\boldsymbol{\theta})$ is strongly convex with respect to the norm $\|\cdot\|_{\boldsymbol{\Sigma}_n}$ at $\boldsymbol{\theta}_*$. This means that there exists $\gamma > 0$ such that

$$\ell_n(\boldsymbol{\theta}_* + \Delta) \geq \ell_n(\boldsymbol{\theta}_*) + \langle \nabla \ell_n(\boldsymbol{\theta}_*), \Delta \rangle + \gamma \|\Delta\|_{\boldsymbol{\Sigma}_n}^2$$

holds for any perturbation $\Delta$ such that $\boldsymbol{\theta}_* + \Delta \in \boldsymbol{\Theta}$. To show this, we derive the Hessian of $\ell_n(\boldsymbol{\theta})$,

$$\nabla^2 \ell_n(\boldsymbol{\theta}) = \frac{1}{n} \sum_{t=1}^n \sum_{j=1}^K \sum_{k=j}^K \sum_{k'=j}^K \frac{\exp[\mathbf{x}_{I_t,\sigma_t(k)}^\top \boldsymbol{\theta} + \mathbf{x}_{I_t,\sigma_t(k')}^\top \boldsymbol{\theta}]}{2 \left( \sum_{\ell=j}^K \exp[\mathbf{x}_{I_t,\sigma_t(\ell)}^\top \boldsymbol{\theta}] \right)^2} \mathbf{z}_{I_t,\sigma_t(k),\sigma_t(k')} \mathbf{z}_{I_t,\sigma_t(k),\sigma_t(k')}^\top \cdot$$

Since $\|\mathbf{x}\| \leq 1$ and $\|\boldsymbol{\theta}\| \leq 1$, we have $\exp[\mathbf{x}^\top \boldsymbol{\theta}] \in [e^{-1}, e]$, and thus

$$\frac{\exp[\mathbf{x}_{I_t,\sigma_t(k)}^\top \boldsymbol{\theta} + \mathbf{x}_{I_t,\sigma_t(k')}^\top \boldsymbol{\theta}]}{\left( \sum_{\ell=j}^K \exp[\mathbf{x}_{I_t,\sigma_t(\ell)}^\top \boldsymbol{\theta}] \right)^2} \geq \frac{e^{-4}}{(K-j)^2} \geq \frac{e^{-4}}{K^2} \geq \frac{e^{-4}}{2K(K-1)} \cdot$$

We further have for any $t \in [n]$ that

$$\sum_{j=1}^K \sum_{k=j}^K \sum_{k'=j}^K \mathbf{z}_{I_t,\sigma_t(k),\sigma_t(k')} \mathbf{z}_{I_t,\sigma_t(k),\sigma_t(k')}^\top \succeq \sum_{k=1}^K \sum_{k'=1}^K \mathbf{z}_{I_t,\sigma_t(k),\sigma_t(k')} \mathbf{z}_{I_t,\sigma_t(k),\sigma_t(k')}^\top$$

$$\succeq 2 \sum_{k=1}^K \sum_{k'=k+1}^K \mathbf{z}_{I_t,k,k'} \mathbf{z}_{I_t,k,k'}^\top \cdot$$

The last step follows from the fact that $\sigma_t$ is a permutation. Simply put, we replace the sum of $K^2$ outer products by all but the ones between the same vectors. Now we combine all claims and get

$$\nabla^2 \ell_n(\boldsymbol{\theta}) \succeq \frac{e^{-4}}{2K(K-1)n} \sum_{t=1}^n \sum_{j=1}^K \sum_{k=j+1}^K \mathbf{z}_{I_t,j,k} \mathbf{z}_{I_t,j,k}^\top = \gamma \boldsymbol{\Sigma}_n$$

for $\gamma = e^{-4}/4$. Therefore, $\ell_n(\boldsymbol{\theta})$ is strongly convex at $\boldsymbol{\theta}_*$ with respect to the norm $\|\cdot\|_{\boldsymbol{\Sigma}_n}$.

**Step 2:** Let $\hat{\boldsymbol{\theta}} = \boldsymbol{\theta}_* + \Delta$. Now we rearrange the strong convexity inequality and get

$$\gamma \|\Delta\|_{\boldsymbol{\Sigma}_n}^2 \leq \ell_n(\boldsymbol{\theta}_* + \Delta) - \ell_n(\boldsymbol{\theta}_*) - \langle \nabla \ell_n(\boldsymbol{\theta}_*), \Delta \rangle \leq -\langle \nabla \ell_n(\boldsymbol{\theta}_*), \Delta \rangle \qquad (18)$$
$$\leq \|\nabla \ell_n(\boldsymbol{\theta}_*)\|_{\boldsymbol{\Sigma}_n^{-1}} \|\Delta\|_{\boldsymbol{\Sigma}_n} \cdot$$

In the second inequality, we use that $\hat{\boldsymbol{\theta}}$ minimizes $\ell_n$, and thus $\ell_n(\boldsymbol{\theta}_* + \Delta) - \ell_n(\boldsymbol{\theta}_*) \leq 0$. In the last inequality, we use the Cauchy-Schwarz inequality.

Next we derive the gradient of the loss function

$$\nabla \ell_n(\boldsymbol{\theta}) = -\frac{1}{n} \sum_{t=1}^n \sum_{j=1}^K \sum_{k=j}^K \frac{\exp[\mathbf{x}_{I_t,\sigma_t(k)}^\top \boldsymbol{\theta}]}{\sum_{\ell=j}^K \exp[\mathbf{x}_{I_t,\sigma_t(\ell)}^\top \boldsymbol{\theta}]} \mathbf{z}_{I_t,\sigma_t(j),\sigma_t(k)} \cdot$$

Zhu et al. [102] note that is a sub-Gaussian random variable and prove that

$$\|\nabla \ell_n(\boldsymbol{\theta}_*)\|_{\boldsymbol{\Sigma}_n^{-1}}^2 \leq \frac{CK^4(d + \log(1/\delta))}{n}$$

holds with probability at least $1 - \delta$, where $C > 0$ is a constant. Finally, we plug the above bound into (18) and get that

$$\gamma \|\Delta\|_{\boldsymbol{\Sigma}_n}^2 \leq \sqrt{\frac{CK^4(d + \log(1/\delta))}{n}} \|\Delta\|_{\boldsymbol{\Sigma}_n}$$

holds with probability at least $1 - \delta$. We rearrange the inequality and since $\gamma$ is a constant,

$$\|\hat{\boldsymbol{\theta}}_n - \boldsymbol{\theta}_*\|_{\boldsymbol{\Sigma}_n}^2 = \|\Delta\|_{\boldsymbol{\Sigma}_n}^2 \leq \frac{CK^4(d + \log(1/\delta))}{n}$$

holds with probability at least $1 - \delta$ for some $C > 0$. This concludes the proof. $\qquad \square$

| Method | Maximum prediction error | Ranking loss |
|---|---|---|
| Dope (ours) | $15.79 \pm 1.08$ | $0.107 \pm 0.002$ |
| Plug-in (400) | $19.75 \pm 1.48$ | $0.104 \pm 0.003$ |
| Plug-in (300) | $30.52 \pm 3.00$ | $0.103 \pm 0.002$ |
| Plug-in (200) | $65.75 \pm 13.71$ | $0.114 \pm 0.003$ |
| Plug-in (100) | $100.39 \pm 10.72$ | $0.142 \pm 0.003$ |
| Optimal | $9.22 \pm 0.82$ | $0.092 \pm 0.002$ |

Table 1: Comparison of Dope with plug-in designs and optimal solution.

## C  Optimal Design for Ranking Feedback

The optimal design for (10) is derived as follows. First, we compute the Hessian of $\ell_n(\boldsymbol{\theta})$,

$$\nabla^2 \ell_n(\boldsymbol{\theta}) = \frac{1}{n} \sum_{t=1}^{n} \sum_{j=1}^{K} \sum_{k=j}^{K} \sum_{k'=j}^{K} \frac{\exp[\mathbf{x}_{I_t,\sigma_t(k)}^{\top}\boldsymbol{\theta} + \mathbf{x}_{I_t,\sigma_t(k')}^{\top}\boldsymbol{\theta}]}{2\left(\sum_{\ell=j}^{K} \exp[\mathbf{x}_{I_t,\sigma_t(\ell)}^{\top}\boldsymbol{\theta}]\right)^2} \mathbf{z}_{I_t,\sigma_t(k),\sigma_t(k')} \mathbf{z}_{I_t,\sigma_t(k),\sigma_t(k')}^{\top} .$$

Its exact optimization is impossible because $\boldsymbol{\theta}_*$ is unknown. This can be addressed in two ways.

**Worst-case design:** Solve an approximation where $\boldsymbol{\theta}_*$-dependent terms are replaced with a lower bound. We take this approach. Specifically, we show in the proof of Lemma 9 that

$$\nabla^2 \ell_n(\boldsymbol{\theta}) \succeq \frac{e^{-4}}{2K(K-1)n} \sum_{t=1}^{n} \sum_{j=1}^{K} \sum_{k=j+1}^{K} \mathbf{z}_{I_t,j,k} \mathbf{z}_{I_t,j,k}^{\top} = \gamma \boldsymbol{\Sigma}_n$$

for $\gamma = e^{-4}/4$. Then we maximize the log determinant of a relaxed problem. This solution is sound and justified, because we maximize a lower bound on the original objective.

**Plug-in design:** Solve an approximation where $\boldsymbol{\theta}_*$ is replaced with a plug-in estimate.

We discuss the pluses and minuses of both approaches next.

**Prior works:** Mason et al. [55] used a plug-in estimate to design a cumulative regret minimization algorithm for logistic bandits. Recent works on preference-based learning [102, 20, 99], which are closest related works, used worst-case designs. Interestingly, Das et al. [20] analyzed an algorithm with a plug-in estimate but empirically evaluated a worst-case design. This indicates that their plug-in design is not practical.

**Ease of implementation:** Worst-case designs are easier to implement. This is because the plug-in estimate does not need to be estimated. Note that this requires solving an exploration problem with additional hyper-parameters, such as the number of exploration rounds for the plug-in estimation.

**Theory:** Worst-case designs can be analyzed similarly to linear models. Plug-in designs require an analysis of how the plug-in estimate concentrates. The logging policy for the plug-in estimate can be non-trivial as well. For instance, the plug-in estimate exploration in Mason et al. [55] is over $\tilde{O}(d)$ special arms, simply to get pessimistic per-arm estimates. Their exploration budget is reported in Table 1. The lowest one, for a 3-dimensional problem, is 6 536 rounds. This is an order of magnitude more than in our Figure 1(b) for a larger 36-dimensional problem.

Based on the above discussion, we believe that worst-case designs strike a good balance between *practicality and theory*. Nevertheless, plug-in designs are appealing because they may perform well with a good plug-in estimate. To investigate this, we repeat Experiment 2 in Section 6 with $K = 2$ (logistic regression). We compare three methods:

**(1)** Dope: This is our method. The exploration policy is $\pi_*$ in (6).

**(2)** Plug-in $(m)$: This is a plug-in baseline. For the first $m$ rounds, it explores using policy $\pi_*$ in (6). After that, we compute the plug-in estimate of $\boldsymbol{\theta}_*$ using (10) and solve the D-optimal design with it. The resulting policy explores for the remaining $n - m$ rounds. Finally, $\boldsymbol{\theta}_*$ is estimated from all feedback using (10).

**(3)** Optimal: The exploration policy $\pi_*$ is computed using the D-optimal design with the unknown $\boldsymbol{\theta}_*$. This validates our implementation and also shows the gap from the optimal solution.

We report both the prediction errors and ranking losses at $n = 500$ rounds in Table 1. The results are averaged over 100 runs. We observe that the prediction error of Dope is always smaller than that of Plug-in (6 times at $m = 100$). Optimal outperforms Dope but cannot be implemented in practice. The gap between Optimal and Plug-in shows that an optimal design with a plug-in estimate of $\boldsymbol{\theta}_*$ can be much worse than that with $\boldsymbol{\theta}_*$. Dope has a comparable ranking loss to Plug-in at $m = 300$ and $m = 400$. Plug-in has a higher ranking loss otherwise.

Based on our discussion and experiments, we do not see any strong evidence to adopt plug-in designs. They would be more complex than worst-case designs, harder to analyze, and we do not see benefits in our experiments. This also follows the principle of Occam's razor, which tells us to design with a minimal needed complexity.

# D   Related Work

The two closest related works are Mehta et al. [56] and Das et al. [20]. They proposed algorithms for learning to rank $L$ pairs of items from pairwise feedback. Their optimized metric is the maximum gap over $L$ pairs. We learn to rank $L$ lists of $K$ items from $K$-way ranking feedback. We bound the maximum prediction error, which is a similar metric to these related works, and the ranking loss in (3), which is novel. Algorithm APO in Das et al. [20] is the closest related algorithmic design. APO greedily minimizes the maximum error in pairwise ranking of $L$ lists of length $K = 2$. Therefore, Dope with ranking feedback (Section 5.1) can be viewed as a generalization of Das et al. [20] to lists of length $K \geq 2$. APO can be compared to Dope by applying it to all possible $\binom{K}{2} L$ lists of length 2 created from our lists of length $K$. We do that in Section 6. The last difference from Das et al. [20] is that they proposed two variants of APO: analyzed and practical. We propose a single algorithm, which is both analyzable and practical.

Our problem can be generally viewed as learning preferences from human feedback [27, 28, 31]. The two most common forms of feedback are pairwise, where the agent observes a preference over two items [12]; and ranking, where the agent observes a ranking of the items [65, 54]. Online learning from human feedback has been studied extensively. In online learning to rank [67, 104], the agent recommends a list of $K$ items and the human provides absolute feedback, in the form of clicks, on all recommended items or their subset. Two popular feedback models are cascading [43, 105, 53] and position-based [44, 23, 100] models. The main difference in Section 4 is that we do not minimize cumulative regret or try to identify the best list of $K$ items. We learn to rank $L$ lists of $K$ items within a budget of $n$ observations. Due to the budget, our work is related to fixed-budget BAI [13, 5, 6, 95]. The main difference is that we do not aim to identify the best arm.

Online learning from preferential feedback has been studied extensively [9] and is often formulated as a dueling bandit [97, 48, 93, 42, 63, 72, 74, 73, 75, 84, 92]. Our work on ranking feedback (Section 5) differs from these works in three main aspects. First, most dueling bandit papers consider pairwise feedback ($K = 2$) while we study a more general setting of $K \geq 2$. Second, a typical objective in dueling bandits is to minimize regret with respect to the best arm, sometimes in context; either in the cumulative or simple regret setting. We do not minimize cumulative or simple regret. We learn to rank $L$ lists of $K$ items. Finally, the acquisition function in dueling bandits is adaptive and updated in each round. Dope is a static design where the exploration policy is precomputed.

Preference-based learning has also been studied in a more general setting of reinforcement learning [88, 60, 94, 29]. Preference-based RL differs from classic RL by learning human preferences through non-numerical rewards [18, 47, 16]. Our work can be also viewed as collecting human feedback for learning policies offline [35, 69, 98, 76]. One of the main challenges of offline learning is insufficient data coverage. We address this by collecting diverse data using optimal designs [66, 24].

Finally, we wanted to compare the ranking loss in (3) to other objectives. There is no reduction to dueling bandits. A classic objective in dueling bandits is to *minimize regret with respect to the best arm* from dueling feedback. Our goal is to *rank $L$ lists*. One could think that our problem can be solved as a contextual dueling bandit, where each list is represented as a context. This is not possible because the context is controlled by the environment. In our setting, the agent controls the chosen list, similarly to APO in Das et al. [20]. Our objective also cannot be reduced to fixed-budget BAI. Our comparisons to Azizi et al. [6] (Sections 4.3 and 5.3) focus on similarities in high-probability bounds. The dependence on $n$ and $d$ is expected to be similar because the probabilities of making a mistake in

| Number of lists $L$ | 100 | 200 | 300 | 400 | 500 | 600 | 700 | 800 |
|---|---|---|---|---|---|---|---|---|
| Time (seconds) | 4.71 | 8.31 | 15.63 | 21.00 | 26.60 | 35.00 | 41.25 | 49.72 |

Table 2: Computation time of policy $\pi_*$ in (6) as a function of the number of lists $L$.

| | $L = 50$ | $L = 100$ | $L = 200$ | $L = 500$ |
|---|---|---|---|---|
| $K = 2$ | $0.12 \pm 0.06$ | $0.28 \pm 0.12$ | $0.37 \pm 0.14$ | $0.57 \pm 0.21$ |
| $K = 3$ | $0.14 \pm 0.06$ | $0.24 \pm 0.10$ | $0.37 \pm 0.15$ | $0.50 \pm 0.19$ |
| $K = 4$ | $0.13 \pm 0.05$ | $0.24 \pm 0.08$ | $0.35 \pm 0.14$ | $0.47 \pm 0.18$ |
| $K = 5$ | $0.12 \pm 0.04$ | $0.21 \pm 0.08$ | $0.34 \pm 0.12$ | $0.45 \pm 0.15$ |

Table 3: The ranking loss of Dope as a function of the number of lists $L$ and items $K$.

our work and Azizi et al. [6] depend on how well the generalization model is estimated, which is the same in both works.

## E  Ablation Studies

We conduct two ablation studies on Experiment 2 in Section 6.

In Table 2, we report the computation time of policy $\pi_*$ in (6). We vary the number of lists $L$ and use CVXPY [21] to solve (6). For $L = 100$, the computation takes 5 seconds; and for $L = 800$, it takes 50. Therefore, it scales roughly linearly with the number of lists $L$.

In Table 3, we vary the number of lists $L$ and items $K$, and report the ranking loss of Dope. We observe that the problems get harder as $L$ increases (more lists to rank) and easier as $K$ increases (longer lists with more feedback).

