# OpenReview forum: "Optimal Design for Human Preference Elicitation"
_NeurIPS.cc/2024/Conference — NeurIPS 2024 poster_

### Official Review · Reviewer_f8gc · 2024-07-06

**Soundness:** 3
**Presentation:** 3
**Contribution:** 3
**Rating:** 5
**Confidence:** 3

**Summary:**

This paper studies the problem of data collection for learning preference models. The key idea is to generalize the optimal design, a method for computing information gathering policies, to ranked lists. The authors study both absolute and relative feedback on the lists.

**Strengths:**

1. The considered problem is useful because collecting human feedback is expensive in practice.

2. The synthetic and real-world experiments show clear advantage of the proposed algorithms while compared with the benchmark methods.

3. This paper is well written and is easy to follow.

**Weaknesses:**

1. One major concern is the use of optimal design in relative feedback setting. In the absolute feedback case in (1), the optimal design used in (5) is correct as it corresponds to maximizing the metric of fisher information matrix in the least squared regression model. However, in the relative feedback case, the likelihood function is based on a multinomial-type distribution (or logistic-type distribution when $K=2$). In this case, the fisher information matrix is different from the one used in the least squared case. The optimal design for generalized linear models is more challenging than that for the linear models. See below the reference.

{\it Stufken, John, and Min Yang. "Optimal designs for generalized linear models." Design and Analysis of Experiments 3 (2012): 137-164.}


2. I found that some assumptions on the features are missing in the main paper. The main paper does not have any assumption on feature vector $x_{I_t, k}$. However, in the proof of Lemma 8 on line 1072 of page 27, the authors said ``(b) follows from independence of $w_s\eta_s$", where $w_s$ and $\eta_s$ are functions of $x$. In addition, the authors claim that they used the G-optimal design result in [48]. However, [48] considered a ridge regression which guarantees the positive definite of the covariate matrix. However, in the proposed Algorithm 1 (line 11), how do you guarantee the positive definiteness of the sample covariance matrix $\Sigma_n$?


3. Assumption 1 assumes the true parameter belong to the constraint parameter space $\Theta$ such that $\theta^T I_d = 0$ and $\|\theta\|_2 \le 1$. Do you require the estimated parameter also belong to this constraint parameter space $\Theta$? I did not see how the estimator in (9) satisfies these two constraints. Does the estimator in line 12 of Algorithm 1 have any scaling issue while compared to the true parameter $\theta^*$ ($\|\theta^*\|_2 \le 1$)?

4. In the Real-world experiment 3 (Nectar dataset) and Real-world experiment 4 (Anthropic dataset), the authors mentioned that ``During simulation, the ranking feedback is generated by the PL model in (2)." This indicates that these experiments are not real-world experiments but still synthetic ones. I would suggest to include some benchmark real-world experiments.


~~~After rebuttal~~~
I have increased the score after discussing with the authors during rebuttal stage.

**Questions:**

see Weakness.

---

> ### Author Rebuttal · Authors · 2024-08-07
>
> We wanted to thank the reviewer for carefully reading the manuscript, and especially for pointing out how to present the algorithms better. We answer all questions below. If you have any additional concerns, please reach out to us to discuss them.
>
> **W1: Optimal design in Section 5**
>
> The design is motivated by prior works [7, 106], which showed that the uncertainty can be represented and optimized using outer products of feature vectors. The advantage of these formulations is that they do not contain model-parameter terms, which appear in the Hessian of the log likelihood. Therefore, the optimal design can be solved similarly to linear models. The additional complexity of GLMs is captured through term $\kappa$ (Assumption 2), which is a lower bound on the derivative of the mean function. This term is common in GLM bandit analyses, although several recent works tried to reduce dependence on it
>
> >> Improved Optimistic Algorithms for Logistic Bandits. ICML 2020.
>
> >> An Experimental Design Approach for Regret Minimization in Logistic Bandits. AAAI 2022.
>
> We leave tightening of the dependence on $\kappa$ in our bounds for future work.
>
> **W2: Feature vector and covariance matrix assumptions**
>
> We assume that the length of feature vectors is at most $1$. This is stated in Assumption 1.
>
> We also want to explain what $w_s$ and $\eta_s$ are. In line 1071 (Lemma 8),
>
> $$
> \underbrace{\mathbf{x}^{\top} \left(\mathbf{X}^{\top} \mathbf{X}\right)^{-1} \mathbf{X}^{\top}}\_{\mathbf{w}^{\top}} \eta
> = \mathbf{w}^\top\eta
> = \sum_{s=1}^t w_s\eta_s$$
>
> where $\eta$ is a vector of independent Gaussian noise up to round $t$. Here $w_s$ and $\eta_s$ are the components of $\mathbf{w}$ and $\eta$. The noise $\eta_s$ is independent Gaussian and does not depend on the feature vector $\mathbf{x}$.
>
> When data are logged according to the optimal design (Chapter 21 of [48]), the sample covariance matrix in the least-squares estimator may not be full rank. This problem can be solved in multiple ways. In Algorithm 12 (Chapter 22 of [48]), each non-zero entry of the logging distribution is rounded up to the closest integer. This yields at most $d (d + 1) / 2$ extra observations. In our experiments, we add $\lambda I_d$, for a small $\lambda > 0$, to line 11 in Algorithm 1. This mostly impacts small sample sizes. Specifically, since the optimal design collects diverse feature vectors, the sample covariance matrix is likely to be full rank when the sample size is large.
>
> **W3: Assumptions on the true and estimated model parameters**
>
> The reviewer is right. In the model parameter estimator in (9), $\arg\min_\theta$ should be replaced with $\arg\min_{\theta \in \Theta}$, where $\Theta$ is defined in Assumption 1. This change should also be done in line 16 of Algorithm 2 (Appendix F). In short, we need the same assumptions as in [106], from which we borrow the estimator and concentration bound.
>
> Our analysis in Section 4 relies on the concentration bound for non-adaptive designs, in (20.3) of [48], which does not depend on the scale of $\theta_*$. Therefore, no assumption on $\theta_*$ is needed in our analysis.
>
> **W4: Semi-synthetic experiments**
>
> The reviewer is right and we will adjust the language. The models in Anthropic and Nectar experiments are learned from real-world data but the feedback is simulated. We need to simulate to collect independent absolute and ranking feedback when the same list of items is explored multiple times.

---

> > ### Comment · Reviewer_f8gc · 2024-08-09
> > **The key issue has not been resolved**
> >
> > My primary concern is that in the case of relative feedback, the likelihood function is based on a multinomial-type distribution. This implies that the Fisher information matrix differs from the one used in least squares scenarios. Specifically, in the relative feedback case, the Fisher information matrix incorporates the derivative of $\mu(x \theta)$, which depends on both $x$ and the unknown parameter $\theta$. This distinction is crucial when comparing the optimal design for Generalized Linear Models to that for linear regression.
> >
> > The authors have also recognized this issue in the response, commenting that "The design is motivated by prior works [7, 106], which showed that the uncertainty can be represented and optimized using outer products of feature vectors. The advantage of these formulations is that they do not contain model-parameter terms, which appear in the Hessian of the log likelihood. Therefore, the optimal design can be solved similarly to linear models. "
> >
> > However, I find the statement "the optimal design can be solved similarly to linear models" to be insufficiently rigorous. Given that the paper focuses on "optimal design" both in its title and content, it is essential for the authors to apply a correct optimal design approach specific to the relative feedback scenario.
> >
> > Indeed, Section 2 of the paper "An Experimental Design Approach for Regret Minimization in Logistic Bandits. AAAI 2022," cited in the authors' response, highlights that "In contrast, for the logistic setting, the G-optimal design objective may be large and we
> > only have a naive bound obtained by naively lower bounding $H(\lambda) \ge \kappa_0 \sum_{x \in {\cal X}} \lambda_x x x^{\top}$. In general these two criteria can produce extremely different designs. We provide an example where these designs are very different in our supplementary, see Figure 3."
> >
> > Given this context, I believe the key concern has not been adequately addressed.

---

> ### Author Response · Authors · 2024-08-10
> **Response to reviewer f8gc**
>
> We thank the reviewer for their response. Our brief response to W1 in the rebuttal was not meant to be "insufficiently rigorous". It goes without saying that we will expand the paper with a discussion of the original problem and our taken approach.
>
> As the reviewer pointed out, our Hessian of the negative log-likelihood is more complex and should be stated. Specifically, let $x_{t, k}$ be the feature vector of the item at position $k$ in the list in round $t$ and $z_{t, k, k'} = x_{t, k} - x_{t, k'}$. Then the Hessian over $n$ rounds is
>
> $$\nabla^2 \ell_n(\theta)
> = \frac{1}{n} \sum_{t = 1}^n
> \sum_{j = 1}^K \sum_{k = j}^K \sum_{k' = j}^K
> \frac{\exp[(x_{t, k} + x_{t, k'})^\top \theta]}
> {2 (\sum_{\ell = j}^K \exp[x_{t, \ell}^\top \theta])^2}
> z_{t, k, k'} z_{t, k, k'}^\top$$
>
> This is shown in lines 1087-1088 in Appendix.
>
> In this work, we maximize the log determinant of relaxed $\nabla^2 \ell_n(\theta_*)$. First, we would like to stress that the exact optimization is impossible unless $\theta_*$ is known. When $\theta_*$ is unknown, two approaches are popular:
>
> 1. A plug-in estimate $\hat{\theta}$ of $\theta_*$ is used. The estimate can be computed by a $\theta$-agnostic optimal design and we discuss it later.
> 2. The $\theta$-dependent term is bounded from below.
>
> We adopt the second approach. Following lines 1089-1090 in Appendix, and using our assumptions that $\\|x_{t, k}\\|_2 \leq 1$ and $\\|\theta\\|_2 \leq 1$, we have
>
> $$\nabla^2 \ell_n(\theta)
> \succeq \frac{e^{- 4}}{2 K (K - 1) n}
> \sum_{t = 1}^n \sum_{j = 1}^K \sum_{k = j + 1}^K
> z_{t, j, k} z_{t, j, k}^\top$$
>
> Therefore, we can maximize the log determinant of relaxed
>
> $$\sum_{t = 1}^n \sum_{j = 1}^K \sum_{k = j + 1}^K
> z_{t, j, k} z_{t, j, k}^\top$$
>
> which we do in algorithm Dope. This solution is sound and justified, because we maximize a lower bound on the original objective.
>
> The last point to discuss is if the alternative approach, maximization of $\nabla^2 \ell_n(\hat{\theta})$ with a plug-in estimate $\hat{\theta}$, could be used. There is no evidence that this approach is practical for the sample sizes of hundreds of human interactions that we consider in our experiments. We start with paper
>
> >> An Experimental Design Approach for Regret Minimization in Logistic Bandits. AAAI 2022.
>
> which you looked at. Note that this paper is for logistic models only, which is a special case of our setting for $K = 2$.
>
> To compute the plug-in estimate, they collect data using optimistic probing in Algorithm 2 based on a G-optimal design for linear models. From the proof of their Theorem 6,
>
> $$O\left(d (\log \log d) \frac{C_0}{\Delta_w^{2}}
> \log\left(\frac{C_0 L}{\Delta_w^2 \delta}\right)\right)$$
>
> samples are needed to compute $\hat{\theta}$, where $L$ is the number of feature vectors and $C_0 \geq 38$. The most favorable setting for this bound is $\Delta_w = 1$ and $C_0 = 38$. Now we instantiate it in our experiments. We take synthetic Experiment 2, where $d = 36$ and $L = 400$. Suppose that we want the claim in Theorem 6 to hold with probability at least $1 - \delta$ for $\delta = 0.05$. Then the suggested sample size is $22043$. This is two orders of magnitude more than our sample sizes in Figure 1b.
>
> While this paper does not contain any experiments with real data, the author report the sample sizes of Algorithm 2 in Table 1. The algorithm collects data to compute the plug-in estimate $\hat{\theta}$ and then collects more data using a $\hat{\theta}$-dependent optimal design to initialize a bandit algorithm. The lowest reported sample size for a $3$-dimensional problem is $6536$. This is an order of magnitude more than our sample sizes in Figure 1b for a larger $36$-dimensional problem.
>
> Another recent work that sheds light on the performance of plug-in and $\theta$-independent optimal designs is
>
> >> Active Preference Optimization for Sample Efficient RLHF. ICML 2024.
>
> The authors analyze an algorithm with plug-in estimates but implement a practical one using the outer product of feature vector differences, similarly to our work. This algorithm is only for the setting of $K = 2$.

---

> > ### Author Response · Authors · 2024-08-13
> >
> > Dear Reviewer f8gc,
> >
> > Can you please let us know if our response addressed your main concern? You argue that we do not apply a correct optimal design approach for the relative feedback scenario. We do, but this was not sufficiently explained in the main paper. All supporting evidence is in Appendix D.1. In summary:
> >
> > 1. The log likelihood of the original problem, its gradient, and its Hessian are all properly stated. See (9) for the log likelihood, line 1097 for the gradient (Appendix D.1), and line 1088 for the Hessian (Appendix D.1). We will bring the Hessian to the main paper and make the relation to the original objective clear.
> > 2. The D-optimal design in the relative feedback setting cannot be solved exactly since the model parameter $\theta_*$ in the Hessian is unknown.
> > 3. To get around this issue in GLMs, a common approach is to eliminate $\theta_*$-dependent terms in the uncertainty model. For instance, the most celebrated works on GLM bandits took this approach. See GLM-UCB and UCB-GLM in
> >
> > >> Parametric Bandits: The Generalized Linear Case. NeurIPS 2010.
> >
> > >> Provably Optimal Algorithms for Generalized Linear Contextual Bandits. ICML 2017.
> >
> > 4. We replace $\theta_*$-dependent terms with a lower bound (line 1089 in Appendix D.1). As a result, the log determinant of the new Hessian is a lower bound on the original one, and its maximization is a sound and justified way of optimizing the original objective.
> > 5. We also discussed another solution, where the lower bounds on $\theta_*$-dependent terms are replaced with a plug-in estimate of $\theta_*$. To the best of our knowledge, there is no evidence that this would result in a better solution than in our paper, at comparable problem and sample sizes. This is discussed in detail in our previous response.
> >
> > Based on the above, we believe that your main concern can be addressed by moving all supporting evidence from Appendix D.1 to the main paper and discussing in detail how the two objectives are related.
> >
> > Sincerely,
> >
> > Authors

---

> > > ### Comment · Reviewer_f8gc · 2024-08-13
> > > **Key issue is not resolved**
> > >
> > > I appreciate the authors for providing additional clarification. However, I would like to further clarify my main concern, which I feel has not been fully addressed. Please correct me if I am wrong.
> > >
> > > 1. As previously mentioned, in the relative feedback scenario, the Hessian of the negative log-likelihood involves an unknown parameter. The authors address this by using a lower bound for one term of the Hessian and then maximizing over the remaining term. To illustrate, consider the problem of solving $\arg\max_x f(x, \theta^*) \times g(x)$. Given that $\theta^*$ is unknown and $f(x, \theta^*) \ge c$, the proposed approach solves $\arg\max_x c \times g(x)$. However, these two solutions may differ significantly because $f(x, \theta^*)$ is itself a function of $x$.
> > >
> > > 2. This issue is also discussed in the paper "An Experimental Design Approach for Regret Minimization in Logistic Bandits" (Mason et al., 2022). In Section 2, the authors state: "In contrast, for the logistic setting, the G-optimal design objective may be large, and we only have a naive bound xxx obtained by naively lower bounding xxx. In general, these two criteria can produce extremely different designs. We provide an example where these designs are very different in our supplementary, see Figure 3."
> > >
> > > 3. Consequently, Mason et al. (2022) suggest estimating $\theta^*$ via $\hat{\theta}$ and then solving $\arg\max_x f(x, \hat{\theta}) \times g(x)$. This approach seems more appropriate to me.
> > >
> > > 4. Furthermore, this method is also recommended in the textbook "Optimal Designs for Generalized Linear Models" as a strategy to address the unknown parameter in the Hessian.
> > >
> > > https://homepages.math.uic.edu/~minyang/research/Stufken%20Yang%20Chap%20Book.pdf
> > >
> > > In summary, I believe it may be more accurate to reconsider using the term 'optimal design' to describe the current approach, particularly in the context of the relative feedback case, where it does not fully align with an 'optimal design' framework. One possible solution could be to tone down the emphasis on 'optimal design' throughout the paper, although this may require some substantial writing revisions.

---

> ### Author Response · Authors · 2024-08-14
>
> We thank the reviewer for their response. The reviewer and us agree that the optimal design problem in Section 5 can be solved in two ways:
>
> 1. **Method A:** Solve an approximation where $\theta_*$-dependent terms are replaced with a lower bound (line 1089 in Appendix D.1). We take this approach.
> 2. **Method B:** Solve an approximation where $\theta_*$ is replaced with a plug-in estimate. The reviewer would like us to take this approach.
>
> Let's examine the pros and cons of both approaches.
>
> **Prior works:** Both the reviewer and us pointed to prior works that took our preferred approaches. Therefore, both approaches can be justified by prior works. Recent works on preference-based learning, which are the closest related works, seem to prefer Method A. For example, see
>
> >> Principled Reinforcement Learning with Human Feedback from Pairwise or K-Wise Comparisons. ICML 2023.
>
> >> Active Preference Optimization for Sample Efficient RLHF. ICML 2024.
>
> >> Provable Reward-Agnostic Preference-Based Reinforcement Learning, ICLR 2024
>
> Interestingly, in the second paper, the authors analyze an algorithm with a plug-in estimate akin to Method B. However, the practical algorithm in experiments uses an approximation akin to Method A. This indicates that Method B may not be practical or yield enough practical benefits.
>
> **Ease of implementation:** Method A is clearly easier to implement. This is because the plug-in estimate in Method B needs to be estimated, which requires solving an additional exploration problem. This also introduces hyper-parameters, such as the number of exploration rounds for the plug-in estimate.
>
> **Theory:** Method A relies on a linear model theory. Method B requires an analysis of how the plug-in estimate concentrates. The logging policy for the plug-in estimate can be quite involved. For instance, the initial exploration in
>
> >> An Experimental Design Approach for Regret Minimization in Logistic Bandits. AAAI 2022.
>
> is over $\tilde{O}(d)$ individual arms, simply to get pessimistic per-arm estimates. The exploration budget is reported in Table 1. The lowest one, for a $3$-dimensional problem, is $6536$. This is an order of magnitude higher budget than in our Figure 1b for a larger $36$-dimensional problem. This indicates that a theoretically-sound design of Method B may be too conservative.
>
> Based on the above discussion, we believe that Method A strikes a good balance between **practicality with a theory support**. We failed to explain in the main paper how the optimal design in Section 5 is approximated. This was an unfortunate omission on our side and we will fix it. All supporting claims are in Appendix D.1 though. We will also stress that the optimal design in Section 5 is not solved optimally. This is not possible and therefore we solve it approximately, as all prior works.
>
> Method B is intriguing because it may perform well with a decent plug-in estimate. The question is whether this would happen within exploration budgets in our paper. To investigate this, we repeat Experiment 2 with $K = 2$ (logistic regression):
>
> * **Dope:** Explore by the policy in (6) for all rounds.
> * **Plug-in ($m$):** Explore by the policy in (6) for $m$ rounds. After that, we compute the plug-in estimate of $\theta_*$ using (9) and solve the D-optimal design with it. This policy explores for the remaining $n - m$ rounds. Finally, $\theta_*$ is estimated from logged data from all rounds.
> * **OPT:** We solve the D-optimal design with $\theta_*$. This validates our implementation and also shows the gap from the optimal solution.
>
> We report both the prediction errors and ranking losses at $n = 500$ rounds. The gap between Dope and Plug-in was larger for $n < 500$. The results are averaged over $100$ runs.
>
> | | Dope (ours)| Plug-in (m = 400) | Plug-in (m = 300) | Plug-in (m = 200) | Plug-in (m = 100) | OPT |
> |-|-|-|-|-|-|-|
> | Maximum prediction error | 15.79 ± 1.08 | 19.75 ± 1.48 | 30.52 ± 3.00 | 65.75 ± 13.71 | 100.39 ± 10.72 | 9.22 ± 0.82 |
> | Ranking loss | 0.107 ± 0.002 | 0.104 ± 0.003 | 0.103 ± 0.002 | 0.114 ± 0.003 | 0.142 ± 0.003 | 0.092 ± 0.002 |
>
> We observe that the prediction error of Dope is always smaller than that of Plug-in ($6$ times at $m = 100$). OPT outperforms Dope but cannot be implemented in practice. The major gap in the performances of OPT and Plug-in shows that an optimal design with a plug-in estimate of $\theta_*$ can perform much worse than with $\theta_*$. Dope has a comparable ranking loss (within margins of error) to Plug-in at $m = 400$ and $m = 300$. Plug-in has a higher ranking loss otherwise. OPT performs the best again.
>
> Based on our discussion and experiments, we do not see any strong evidence for why we should adopt Method B. It would be more complex than Method A, harder to analyze, and we do not see benefits in our experiments. This also follows the principle of Occam's razor, which tells us to design with a minimal needed complexity.

---

### Official Review · Reviewer_sgSH · 2024-07-16

**Soundness:** 4
**Presentation:** 4
**Contribution:** 3
**Rating:** 7
**Confidence:** 3

**Summary:**

The paper considers experiment design for collecting ranking/direct feedback, with an application to fine tuning language models.
Author formulate active exploration for fine-tuning as a ranking problem and propose an algorithm which satisfies the standard guarantees for its ranking loss. Experiments on synthetic and real data are provided as a proof of concept, which show that optimal design consistently outperforms random sampling and a number of other vanilla baselines.

**Strengths:**

- The problem setting applies well to the proposed RLHF/AIHF application. Although this is not used in sota models, but I think the ranking loss gives a strong criteria for fine-tuning.

- The paper is well written. All relevant theoretical guarantees for understanding the algorithm are provided and the rates are compared to previous results.

- Authors provide proof of concept experiments on a large scale AIHF dataset which almost exactly matches their problem setting and demonstrate the benefits of using optimal design.

I think one rarely comes across a paper that manages to deliver on all these criteria.

**Weaknesses:**

- From a purely theoretical standpoint, I am not sure how novel is the result. I am not particularly familiar with the literature on optimal design. However under slightly different problem setting (e.g. BAI for linear bandits), and the complexity bounds are pretty much common knowledge, even with dueling (k-wise or pair-wise) feedback.

- Some (recent) related work on dueling bandits with function approximation (linear, kernelized, admissible setting) are missing. Particularly [1] proposes a very similar approach: active exploration by maximizing the $\log\mathrm{det} V_t$ and MLE for estimation. I put the ones I could think of below.

- I think comparing to other algorithms (not just toy baselines) would really strengthen the story and relevance of the paper. Majority (if not all) of the baselines consider dueling regret (or a sub-optimality gap based on that) but can still be used to learn the reward model for the Nectar or the HH experiment [see 1 & 4]. The data/collection model would be different (e.g. no Lists) but the algorithms can still be evaluated based on the ranking loss to make them comparable to Dope. I think the current way of instantiating a dueling design withing the ranking framework is not realistic.

[1] Nirjhar Das, Souradip Chakraborty, Aldo Pacchiano and Sayak Ray Chowdhury Active Preference Optimization for Sample Efficient RLHF. arXiv preprint, 2024.

[2] Barna Pásztor, Parnian Kassraie, and Andreas Krause. Bandits with Preference Feedback: A Stackelberg Game Perspective. arXiv preprint, 2024.

[3] Johannes Kirschner and Andreas Krause. Bias-robust bayesian optimization via dueling bandits. In International Conference on Machine Learning. PMLR, 2021.

[4] Viraj Mehta, Vikramjeet Das, Ojash Neopane, Yijia Dai, Ilija Bogunovic, Jeff Schneider, and Willie Neiswanger. Sample efficient reinforcement learning from human feedback via active exploration. arXiv preprint, 2023a.

[5] Viraj Mehta, Ojash Neopane, Vikramjeet Das, Sen Lin, Jeff Schneider, and Willie Neiswanger. Kernelized offline contextual dueling bandits. arXiv preprint, 2023b.

[6] Aadirupa Saha. Optimal algorithms for stochastic contextual preference bandits. Advances in Neural Information Processing Systems, 34:30050–30062, 2021

[7] Aadirupa Saha, and Akshay Krishnamurthy. "Efficient and optimal algorithms for contextual dueling bandits under realizability." International Conference on Algorithmic Learning Theory. PMLR, 2022.

[8] Shion Takeno, Masahiro Nomura, and Masayuki Karasuyama. Towards practical preferential bayesian optimization with skew gaussian processes. In International Conference on Machine Learning, pages 33516–33533. PMLR, 2023

[9] Yichong Xu, Aparna Joshi, Aarti Singh, and Artur Dubrawski. Zeroth order non-convex optimization with dueling-choice bandits. In Conference on Uncertainty in Artificial Intelligence. PMLR, 2020

[10] Wenjie Xu, Wenbin Wang, Yuning Jiang, Bratislav Svetozarevic, and Colin N Jones. Principled preferential bayesian optimization. arXiv preprint arXiv:2402.05367, 2024.

**Questions:**

1. How does the problem of ranking L lists compare to a finite arm (linear) dueling bandit problem with $k$-wise feedback? Is there a reduction from one to another?
2. How does the ranking problem compare to best-arm-identification for contextual bandits? I'm curious if one can formally show an equivalence between the ranking loss and the BAI sub-optimality gap [e.g. in 1, 2]?
3. Is dependence on $\kappa$ in Theorem 6 improvable? I can imagine that $\kappa$ can get really small exponentially fast?


[1] Das, Nirjhar, et al. "Provably sample efficient rlhf via active preference optimization." arXiv preprint arXiv:2402.10500 (2024).
[2] Azizi, Mohammad Javad, Branislav Kveton, and Mohammad Ghavamzadeh. "Fixed-budget best-arm identification in structured bandits." arXiv preprint arXiv:2106.04763 (2021).

**Limitations:**

The limitations are discussed.

---

> ### Author Rebuttal · Authors · 2024-08-07
>
> We thank the reviewer for detailed feedback and positive evaluation of the paper. We answer all questions below. If you have any additional concerns, please reach out to us to discuss them.
>
> **W1: Novelty in optimal designs**
>
> A good introduction to optimal designs is Chapter 21 in [48]. At a high level, optimal designs are a tool for computing optimal uncertainty reduction policies. The policies are non-adaptive and thus can be precomputed, which is one of their advantages. Adaptive bandit algorithms can be obtained by combining optimal designs and elimination. A good example is Chapter 22 in [48]. Therefore, solving an optimal design opens the door to other solutions. To the best of our knowledge, this is the first work to study optimal designs in ranking problems, from both absolute and ranking feedback.
>
> **W2: Comparison to related works on dueling bandits**
>
> Thank you for the numerous [1-10] references to dueling bandits. To simplify comparison, we focus on three main differences:
>
> **$K = 2$ versus $K > 2$:** All of [1-10] are dueling bandit papers ($K = 2$) while we study a more general setting of $K \geq 2$.
>
> **Worst-case optimization over lists:** A classic objective in dueling bandits is to *minimize regret with respect to the best arm* from dueling feedback, sometimes in context. This problem can be studied in both cumulative and simple regret settings. The papers [2-3] and [5-10] are of this type. Our goal is to *sort $L$ lists* and the agent controls the chosen list. The works [1] and [4] are closest to our work in this aspect.
>
> **Adaptive versus static design:** All of [1-10] are adaptive designs, where the acquisition function is updated in each round. Dope is a static design where the exploration policy is precomputed. Interestingly, the practical variant of APO in [1] can also be viewed as a static design, because the optimized covariance matrix depends only on the feature vectors of observed lists but not the observations.
>
> **Q1: Reduction of ranking $L$ lists to dueling bandits**
>
> There is no reduction because the objectives are different. A classic objective in dueling bandits is to *minimize regret with respect to the best arm* from dueling feedback. Our goal is to *sort $L$ lists*. One may think that our problem could be solved as a contextual dueling bandit, where each list is represented as context. This is not possible because the context is controlled by the environment. In our setting, the agent controls the chosen list, similarly to APO in [1].
>
> **Q2: Equivalence of objectives with [1] and [2]**
>
> Algorithm APO in [1] is indeed the closest related work and we wanted to thank you for bringing it up. APO greedily minimizes the maximum error in pairwise ranking of $L$ lists of length $K = 2$. Therefore, it can be applied to our setting by applying it to all possible ${K \choose 2} L$ lists of length $2$ created from our lists of length $K$, as described in lines 296-300. We compare to APO next, both empirically and algorithmically.
>
> **Empirical comparison:** We first report the ranking loss in Experiment 2 (Figure 1b) where $K = 4$:
>
> | | n = 10 | n = 20 | n = 50 | n = 100 |
> |-|--------|--------|--------|---------|
> | Dope (ours) | 1.1 ± 0.049 | 0.78 ± 0.029 | 0.48 ± 0.017 | 0.32 ± 0.010 |
> | APO | 1.5 ± 0.057 | 0.99 ± 0.037 | 0.62 ± 0.022 | 0.48 ± 0.021 |
>
> Next we report the ranking loss on the Nectar dataset (Figure 1c) where $K = 5$:
>
> | | n = 50 | n = 100 | n = 200 | n = 500 |
> |-|--------|---------|---------|---------|
> | Dope (ours) | 0.51 ± 0.066 | 0.40 ± 0.053 | 0.29 ± 0.038 | 0.19 ± 0.027 |
> | APO | 1.00 ± 0.120 | 0.98 ± 0.110 | 0.75 ± 0.095 | 0.73 ± 0.100 |
>
> We observe that Dope has a significantly lower ranking loss than APO. This is for two reasons. First, APO solves the uncertainty reduction problem greedily. Dope solves it optimally, by sampling from an optimal design. Second, APO is designed for $K = 2$ items per list. While it can be applied to our problem, it is suboptimal because it does not leverage the $K$-way feedback that Dope uses.
>
> **Algorithmic comparison:** Dope with ranking feedback (Section 5) can be viewed as a generalization of [1] to lists of length $K \geq 2$. [1] propose 2 methods: one is analyzed and one is practical. We propose a single algorithm, which is practical and analyzable; and provide both prediction error (Theorem 5) and ranking (Theorem 6) guarantees.
>
> There is no equivalence of objectives with [2]. Our discussion in lines 207-214 focuses on similarites in high-probability bounds. The dependence on $n$ and $d$ is expected to be similar because the probability of making a mistake in [2] or a ranking error in our work depends on how well the generalization model is estimated, which is the same in both works.
>
> **Q3: Dependence on $\kappa$ in Theorem 6**
>
> We briefly discuss this in lines 270-275. Theorem 6 is similar to existing sample complexity bounds for fixed-budget BAI in GLMs. In these bounds, the additional complexity of GLMs is captured through term $\kappa$ (Assumption 2), which is a lower bound on the derivative of the mean function. This term is common in GLM bandit analyses, although several recent works tried to reduce dependence on it
>
> >> Improved Optimistic Algorithms for Logistic Bandits. ICML 2020.
>
> >> An Experimental Design Approach for Regret Minimization in Logistic Bandits. AAAI 2022.
>
> We leave tightening of the dependence on $\kappa$ in our bounds for future work.

---

> > ### Comment · Reviewer_sgSH · 2024-08-13
> >
> > Thank you for your response and particularly comparison with APO.
> >
> > I just want to point out the following. I think the dueling bandit setting can be extended to $K$ arm in a less trivial way: by writing the categorical likelihood function and keeping $L$ lists, instead of increasing the number of lists and making them of length $2$. So I am not entirely sure if this is the most fair comparison of the algorithms, but it is indeed still insightful to demonstrate the benefits of $K$-wise vs pairwise feedback in modeling the problem.
> >
> > Overall, I recommend the paper for acceptance: it delivers on theory, methodology, and real-world experiments. Further, the problem setting is relevant to the considered applications and active fine-tuning of LLMs, making up a truly strong work.

---

> > > ### Author Response · Authors · 2024-08-13
> > >
> > > Thank you for the response and supporting our work!

---

### Official Review · Reviewer_Chgi · 2024-07-21

**Soundness:** 3
**Presentation:** 3
**Contribution:** 2
**Rating:** 5
**Confidence:** 3

**Summary:**

This manuscript deals with preference models which are at the crossroads of linear bandits, ranking models, and optimal designs. The authors consider a model where one has to rank L lists of K objects. The expected reward for object k of list i is x_{i,k}\theta^* and \theta^* \in \mathbb{R}^d is some unknown quantity. They consider two feedback models:  (i) the "absolute feedback model" where the learners observes the noisy reward and (ii) the "ranking feedback model" where the learner observes an order of the K items sampled from the celebrated Plackett-Luce model.  In the first model, the authors introduce an optimal design with total budget n for estimating the parameter \theta^* and plug the estimator to bound the error in estimating the L ranking of the K objects. Then, they extend their procedure to the ranking feedback model.

**Strengths:**

1) This manuscript introduces an extension of Kiefer-Wolfowitz theorem for building an optimal design in maximum prediction error in linear regression. Although the proof ideas are quite similar to classical Kiefer-Wolfowitz theorem, this result seems to be new.

2) This allows them  control both the prediction error and the ranking error under both feedback models. Both errors seem to be order-wise optimal.

3) The virtue of their procedure compared to adaptive one is that both the experiment design and the computation of the estimators are pretty simple.

4) The authors illustrate the benefits of their procedure compared to e.g. uniform design on small-scale synthetic numerical experiments.

**Weaknesses:**

1) The purpose of building G-optimal lemmas is to derive tight optimal bounds, However, in this manuscript (Theorems 3--6), the authors only provide order-wise optimal bounds.

1-a) In Theorem 3, the authors establish that the maximum prediction error is of order (up to log) d^2/n. However, for obtaining such a bound, there is no need  to rely on optimal design. Simply using some variant of a uniform design would be sufficient.
1-b) In Theorem 4, the authors plug the analysis of the OLS for the ranking purpose. However, there is no reason why a G-optimal design should be optimal ranking. Indeed, the ranking loss is not a linear function of the prediction loss. Optimizing the ranking error could therefore require quite different designs. For instance, if two items are extremely difficult to compare, then an optimal ranking design should put further emphasis on this comparison than an optimal prediction design.
1-c) In generalized linear models, solving the problem (6) does not necessary lead to an optimal prediction design. Still, the author use this approach in Section 5 for ranking feedback. Hence, it is not clear to what extent this design is the most relevant for the prediction purpose.


2) Minor remark: Lemma 2 and Theorem 5 require that n  is larger than the number of lists L. This seems to be in contradiction with the regime L >n which is put forward in Section 2.

**Questions:**

1) It is not clear to me to what extent the preference models considered in this manuscript have been previously studied. If it is case, could the authors further discuss this literature? If it is not the case, could the authors explain why the linear assumption with observed x_{i,k} is realistic?

2) It is not clear to me to what extent the rate d^2/n is optimal in Theorem 3. Could the authors discuss this rate?

**Limitations:**

As explained in the weakness section, I feel that the limitations of using the G-optimal design for ranking purposes are not discussed enough in the manuscript.

---

> ### Author Rebuttal · Authors · 2024-08-07
>
> We thank the reviewer for detailed feedback. We answer all questions below. If you have any additional concerns, please reach out to us to discuss them.
>
> **W1a: Uniform design would suffice to get a $\tilde{O}(d^2 / n)$ rate in Theorem 3**
>
> We respectfully disagree. Consider the following example. Take $K = 2$. Let $x_{i, 1} = (1, 0, 0)$ for $i \in [L - 1]$ and $x_{L, 1} = (0, 1, 0)$, and $x_{i, 2} = (0, 0, 1)$ for all $i \in [L]$. In this case, the minimum eigenvalue of $\bar{\Sigma}_n$ is $n / L$ is expectation, because only one item in list $L$ provides information about the second feature $(0, 1, 0)$. Following the same steps as in Theorem 3, we would get a rate of $\tilde{O}(d L / n)$. A similar observation was also made in prior works on optimal designs, such as [25] and
>
> >> Best-Arm Identification in Linear Bandits. NeurIPS 2014.
>
> **W1b: Is the optimal design in Section 4 optimal for ranking?**
>
> We agree with the reviewer that our optimal design may not be optimal for ranking. This is because our ranking bound in Theorem 4 is derived using a prediction error bound. We have not focused solely on the optimal design for ranking because we see value in both prediction (Theorem 3) and ranking (Theorem 4) bounds. The fact that we provide both shows the versatility of our approach. We discuss the tightness of the bound in Theorem 4 after the claim. The dependence on $n$ and $d$ is similar to prior works on fixed-budget BAI in linear models and likely optimal. This is because their probability of making a mistake or a ranking error in our work depends on how well the linear model is estimated, which is the same in both works.
>
> **W1c: Optimal design in Section 5**
>
> The design is motivated by prior works [7, 106], which showed that the uncertainty can be represented and optimized using outer products of feature vectors. The advantage of these formulations is that they do not contain model-parameter terms, which appear in the Hessian of the log likelihood. Therefore, the optimal design can be solved similarly to linear models. The additional complexity of GLMs is captured through term $\kappa$ (Assumption 2), which is a lower bound on the derivative of the mean function. This term is common in GLM bandit analyses, although several recent works tried to reduce dependence on it
>
> >> Improved Optimistic Algorithms for Logistic Bandits. ICML 2020.
>
> >> An Experimental Design Approach for Regret Minimization in Logistic Bandits. AAAI 2022.
>
> We leave tightening of the dependence on $\kappa$ in our bounds for future work.
>
> **W2: Lemma 2 and Theorem 5 require that $n > L$**
>
> We do not think so. Can the reviewer be more specific? Both claims contain maximization over lists, $\max_{i \in [L]}$, and not a summation, $\sum_{i = 1}^L$. While the optimized covariance matrix $V_\pi$ (line 130) involves $\sum_{i = 1}^L$, the $i$-th term is weighted by the probability that list $i$ is chosen. By the Kiefer-Wolfowitz theorem, the probability distribution that solves the problem is sparse, has at most $d (d + 1) / 2$ non-zero entries.
>
> **Q1: Have our preference models been studied before?**
>
> Yes. The absolute feedback model is a variant of a click model (line 92). Click models have been studied in contextual bandits since [110] and
>
> >> Contextual Combinatorial Cascading Bandits. ICML 2016.
>
> The probability of a click in these papers is a linear function of context and an unknown model parameter. The ranking feedback model is the same as in [106].
>
> **Q2: Optimality of $\tilde{O}(d^2 / n)$ rate in Theorem 3**
>
> A high-probability upper bound on the prediction error in linear models under the optimal design is
>
> $$|x^\top (\hat{\theta}\_n - \theta\_{\star})|
> \leq \underbrace{||\hat{\theta}\_n - \theta_*|\|_{\bar{\Sigma}_n}}\_{\tilde{O}(\sqrt{d})} \cdot
> \underbrace{||x||\_{\bar{\Sigma}_n^{-1}}}\_{\tilde{O}(\sqrt{d / n})}
> = \tilde{O}(d / \sqrt{n})$$
>
> This bound can be derived by combining (20.3) and Claim 3 in Theorem 21.1, both in [48]. The square of the prediction error is bounded as $(x^\top (\hat{\theta}\_n - \theta\_{\star}))^2 = \tilde{O}(d^2 / n)$. The same rate is achieved in Theorem 3. This is optimal since we bound the squared prediction error for $K$ items ($K$ times more than in a classic linear model) from batches of $K$ observations ($K$ times faster learning). The classic setting can be also viewed as $K = 1$.

---

> > ### Comment · Reviewer_Chgi · 2024-08-08
> >
> > Thank you for your response. Regarding the problems in the lemmas, you assume  in lemma 2 that n pi*(i) is an integer. This is not possible unless n>L. This lemma is used in the proof of theorem 5.

---

> > > ### Author Response · Authors · 2024-08-08
> > > **Response to Reviewer Chgi**
> > >
> > > We thank the reviewer for the quick response. We answer the question below.
> > >
> > > Your conclusion would be correct if all entries of $\pi_*$ could be non-zero. However, we prove in Theorem 1 (Matrix Kiefer-Wolfowitz) that $\pi_*$ has at most $d (d + 1) / 2$ non-zero entries (line 142). Note that this is independent of the number of lists $L$. In fact, the claim would also hold for infinitely many lists, similarly to Chapter 21.1 in [48].
> > >
> > > A natural question to ask is if the integer condition in Lemma 2 could be further relaxed. The answer is yes and we will comment on this in the next version of the paper. The key idea is to round each non-zero entry of $n \pi_*(i)$ up to the closest integer. As an example, if $n \pi_*(i)$ was $3.7$, the number of observations of list $i$ would be $4$. This will clearly result in an integer allocation of size at most $n + d (d + 1) / 2$. All our current claims would hold for any $\pi_*$ and this allocation.

---

> > > > ### Author Response · Authors · 2024-08-13
> > > >
> > > > Dear Reviewer Chgi,
> > > >
> > > > Can you please let us know if our response addressed your last remaining concern?
> > > >
> > > > In summary, the optimal design solution $\pi_*$ has at most $d (d + 1) / 2$ non-zero entries. Therefore, having a budget $n$ such that all $n \pi_*(i)$ are integers does not require a number of observations $n$ that is proportional to the number of lists $L$. In the same response, we also suggest a rounding procedure that relaxes the integer condition.
> > > >
> > > > Sincerely,
> > > >
> > > > Authors

---

> > > > > ### Comment · Reviewer_Chgi · 2024-08-13
> > > > >
> > > > > Thank you, this adresses this minor comment. I home that à final version will provide suitable proofs that deal with this general case.
> > > > >
> > > > > Stiller, you did not answer the main criticism in my review (order wise bounds  vs tight optimal). Besides, in your response about optimality of the rate, you do not explain why and in which sense the rate cannot be faster than d^2/n.

---

> ### Author Response · Authors · 2024-08-13
> **Response to Reviewer**
>
> Thank you for getting back with additional questions.
>
> **Tightness of the bounds**
>
> We agree with the reviewer that our optimal designs may not be optimal for ranking. We have not focused solely on the optimal design for ranking because we see value in both prediction (Theorem 3) and ranking (Theorem 4) bounds. The fact that we provide both shows the versatility of our approach.
>
> We also wanted to comment on our solution to the optimal design problem in Section 5. We minimize the prediction error by approximately solving the problem. Specifically:
>
> 1. The log likelihood of the original problem, its gradient, and its Hessian are all stated at the following places: (9) for the log likelihood, line 1097 for the gradient (Appendix D.1), and line 1088 for the Hessian (Appendix D.1).
> 2. The D-optimal design in the relative feedback setting cannot be solved exactly since the model parameter $\theta_*$ in the Hessian is unknown.
> 3. To get around this issue in GLMs, a common approach is to eliminate $\theta_*$-dependent terms in the uncertainty model. Many works using GLMs in decision making took this approach,
>
> >> Parametric Bandits: The Generalized Linear Case. NeurIPS 2010.
>
> >> Provably Optimal Algorithms for Generalized Linear Contextual Bandits. ICML 2017.
>
> >> Principled Reinforcement Learning with Human Feedback from Pairwise or K-Wise Comparisons. ICML 2023.
>
> >> Active Preference Optimization for Sample Efficient RLHF. ICML 2024.
>
> >> Provable Reward-Agnostic Preference-Based Reinforcement Learning, ICLR 2024
>
> 4. We replace $\theta_*$-dependent terms with a lower bound (line 1089 in Appendix D.1). As a result, the log determinant of the new Hessian is a lower bound on the original one, and its maximization is a sound and justified way of optimizing the original objective.
> 5. The penalty for solving the optimal design approximately appears in our claims as a constant of roughly $5$. This is because the norms of $\theta_*$ and feature vectors are all bounded by $1$. We will make this constant explicit in the next version of the paper.
>
> **Optimality of $\tilde{O}(d^2 / n)$ rate in Theorem 3**
>
> An upper bound on the prediction error in the linear model, when proved through the Cauchy–Schwarz inequality, is
>
> $$|x^\top (\hat{\theta}\_n - \theta\_{\star})|
> \leq \underbrace{||\hat{\theta}\_n - \theta\_{\star}||\_{\bar{\Sigma}\_n}}\_{\tilde{O}(\sqrt{d})} \cdot
> \underbrace{||x||\_{\bar{\Sigma}\_n^{-1}}}\_{\tilde{O}(\sqrt{d / n})}
> = \tilde{O}(d / \sqrt{n})$$
>
> with a high probability. This bound holds for infinitely many feature vectors. The bound can be tightened for a finite number of feature vectors, $m$, to $\tilde{O}(\sqrt{d / n})$, where the $\tilde{O}$ hides $\sqrt{\log m}$. This can be proved using a union bound over (20.3) in Chapter 20 of [48]. When bounding the square of the error, the above bounds become $\tilde{O}(d^2 / n)$ and $\tilde{O}(d / n)$, where the second $\tilde{O}$ hides $\log m$.
>
> In Theorem 3, we show that the prediction error is $\tilde{O}(d^2 / n)$. This matches the rate in the linear model and holds for infinitely many lists. The extra factor of $d$ can be eliminated by assuming a finite number of lists.

---

### Official Review · Reviewer_HJve · 2024-07-30

**Soundness:** 4
**Presentation:** 3
**Contribution:** 3
**Rating:** 7
**Confidence:** 3

**Summary:**

This paper presents a novel approach for data collection to learn preference models from human feedback. The key innovation is generalizing optimal design, a method for computing information gathering policies, to ranked lists. The authors study both absolute and relative feedback settings, developing efficient algorithms for each and providing theoretical analyses. They prove that their preference model estimators improve with more data, as does the ranking error under these estimators. The work is evaluated on several synthetic and real-world datasets, demonstrating the statistical efficiency of the proposed algorithms. This research contributes to the field by providing a theoretically grounded and empirically effective method for learning preference models from human feedback, with potential applications in areas such as reinforcement learning from human feedback (RLHF) and information retrieval.

**Strengths:**

The paper extends optimal design theory to ranked lists, introducing a novel approach to preference learning. This is evidenced by the generalization of the Kiefer-Wolfowitz theorem to matrices, providing a strong theoretical foundation for the Dope algorithm.

**Weaknesses:**

I believe this is a solid piece of work with no obvious errors. From my perspective, considering the following points could potentially strengthen the paper:
1. In the synthetic experiments (Figures 1a and 1b), the authors only consider the case of L=400 and K=4. Authors should consider different combinations of L and K values to provide a more comprehensive evaluation of the algorithm's performance. In particular, testing cases where K>4 could reveal potential issues arising from the K^6 term in the theoretical analysis.
2. The paper compares the proposed Dope algorithm to several baselines, but it lacks comparison to more recent and sophisticated methods in preference learning and active learning.

**Questions:**

No.

**Limitations:**

No.

---

> ### Author Rebuttal · Authors · 2024-08-07
>
> We thank the reviewer for positive feedback and acknowledging that our work is solid. We answer all questions below. If you have any additional concerns, please reach out to us to discuss them.
>
> **Q1: Synthetic experiments beyond $L = 400$ and $K = 4$**
>
> We vary the number of lists and items, $L \in \\{50, 100, 200, 500\\}$ and $K \in \\{2, 3, 4, 5\\}$, and report the ranking loss of Dope in Experiment 2 (Figure 1b):
>
> | | L = 50 | L = 100 | L = 200 | L = 500 |
> |-|--------|---------|---------|---------|
> | K = 2 | 0.12 ± 0.06 | 0.28 ± 0.12 | 0.37 ± 0.14 | 0.57 ± 0.21 |
> | K = 3 | 0.14 ± 0.06 | 0.24 ± 0.10 | 0.37 ± 0.15 | 0.50 ± 0.19 |
> | K = 4 | 0.13 ± 0.05 | 0.24 ± 0.08 | 0.35 ± 0.14 | 0.47 ± 0.18 |
> | K = 5 | 0.12 ± 0.04 | 0.21 ± 0.08 | 0.34 ± 0.12 | 0.45 ± 0.15 |
>
> We observe that the problems get harder as $L$ increases (more lists to rank) and easier as $K$ increases (longer lists but also more feedback).
>
> **Q2: Empirical comparison to a recent baseline**
>
> We compare Dope to a state-of-the-art algorithm APO from
>
> >> Active Preference Optimization for Sample Efficient RLHF. ICML 2024.
>
> This recent work was suggested by **Reviewer sgSH** and is the closest related work. APO greedily minimizes the maximum error in pairwise ranking of $L$ lists of length $K = 2$. We extend it to $K > 2$ by applying it to all possible ${K \choose 2} L$ lists of length $2$ created from our lists of length $K$, as described in lines 296-300. We first report the ranking loss in Experiment 2 (Figure 1b) where $K = 4$:
>
> | | n = 10 | n = 20 | n = 50 | n = 100 |
> |-|--------|--------|--------|---------|
> | Dope (ours) | 1.1 ± 0.049 | 0.78 ± 0.029 | 0.48 ± 0.017 | 0.32 ± 0.010 |
> | APO | 1.5 ± 0.057 | 0.99 ± 0.037 | 0.62 ± 0.022 | 0.48 ± 0.021 |
>
> Next we report the ranking loss on the Nectar dataset (Figure 1c) where $K = 5$:
>
> | | n = 50 | n = 100 | n = 200 | n = 500 |
> |-|--------|---------|---------|---------|
> | Dope (ours) | 0.51 ± 0.066 | 0.40 ± 0.053 | 0.29 ± 0.038 | 0.19 ± 0.027 |
> | APO | 1.00 ± 0.120 | 0.98 ± 0.110 | 0.75 ± 0.095 | 0.73 ± 0.100 |
>
> We observe that Dope has a significantly lower ranking loss than APO. This is for two reasons. First, APO solves the uncertainty reduction problem greedily. Dope solves it optimally, by sampling from an optimal design. Second, APO is designed for $K = 2$ items per list. While it can be applied to our problem, it is suboptimal because it does not leverage the $K$-way feedback that Dope uses.
>
> We will include all new experiments in the next version of the paper.

---

### Author Response · Authors · 2024-08-12
**Gentle Reminder to the Reviewers**

Dear reviewers,

Thank you for the reviews and taking our rebuttal into account when evaluating the paper. The author-reviewer discussion will end soon (August 13 EoD AoE). If you have any additional questions or concerns, stemming from our rebuttal or the follow-up explanations, we would be happy to answer them in the next 2 days.

Sincerely,

Authors

---

### Decision · Program_Chairs · 2024-09-25

**Decision:**

Accept (poster)

**Comment:**

This paper studies the problem of data collection for learning preference models. The key idea is to generalize the optimal design, a method for computing information gathering policies, to ranked lists. The authors study both absolute and relative feedback on the lists.

This paper is borderline, but more on the accept side, based on the reviews and ratings. All reviewers have recommended to accept this paper.
Also, my understanding is that the authors have addressed most of concerns raised by the reviewers. This paper also has many strengths, as have been pointed out by the reviewers.

Finally, I agree with Reviewer f8gc that this paper should reduce the emphasis on "optimal design" and consider using terms like "active learning" or something similar.